# Neuropeptide B mediates female sexual receptivity in medaka fish, acting in a female-specific but reversible manner

Towako Hiraki-Kajiyama[1,2], Junpei Yamashita[1], Keiko Yokoyama[1], Yukiko Kikuchi[1], Mikoto Nakajo[1,3], Daichi Miyazoe[1], Yuji Nishiike[1], Kaito Ishikawa[1], Kohei Hosono[1], Yukika Kawabata-Sakata[1,4], Satoshi Ansai[5,6], Masato Kinoshita[5], Yoshitaka Nagahama[7], Kataaki Okubo[1]*

[1]Department of Aquatic Bioscience, Graduate School of Agricultural and Life Sciences, The University of Tokyo, Tokyo, Japan; [2]Laboratory for Systems Molecular Ethology, RIKEN Center for Brain Science, Wako, Japan; [3]Department of Biological Sciences, Graduate School of Science, The University of Tokyo, Tokyo, Japan; [4]Department of Pathophysiology, Tokyo Medical University, Tokyo, Japan; [5]Division of Applied Biosciences, Graduate School of Agriculture, Kyoto University, Kyoto, Japan; [6]Laboratory of Bioresources, National Institute for Basic Biology, Okazaki, Japan; [7]Division of Reproductive Biology, National Institute for Basic Biology, Okazaki, Japan

**Abstract** Male and female animals display innate sex-specific mating behaviors. In teleost fish, altering the adult sex steroid milieu can effectively reverse sex-typical mating behaviors, suggesting remarkable sexual lability of their brains as adults. In the teleost medaka, neuropeptide B (NPB) is expressed female-specifically in the brain nuclei implicated in mating behavior. Here, we demonstrate that NPB is a direct mediator of estrogen action on female mating behavior, acting in a female-specific but reversible manner. Analysis of regulatory mechanisms revealed that the female-specific expression of NPB is dependent on direct transcriptional activation by estrogen via an estrogen-responsive element and is reversed in response to changes in the adult sex steroid milieu. Behavioral studies of NPB knockouts revealed that female-specific NBP mediates female receptivity to male courtship. The female-specific NPB signaling identified herein is presumably a critical element of the neural circuitry underlying sexual dimorphism and lability of mating behaviors in teleosts.

DOI: https://doi.org/10.7554/eLife.39495.001

*For correspondence:
okubo@marine.fs.a.u-tokyo.ac.jp

Competing interests: The authors declare that no competing interests exist.

## Introduction

From invertebrates to humans, males and females of a given species exhibit profound differences in mating behavior; in general, males perform elaborate courtship displays to attract females for mating, and females evaluate male courtship displays to decide whether to mate. Such differences in mating behavior between males and females result from sexually dimorphic development and activation of the underlying neural circuits (*Yang and Shah, 2016*). In vertebrates, sex-typical mating behaviors and the underlying neural circuits are highly dependent on the sex steroid hormone milieu, which is unique to each sex (*McCarthy and Arnold, 2011*; *McCarthy et al., 2017*). Males have higher circulating levels of androgen derived from the testis, whereas females have a cyclic pattern of estrogen and progestin arising from the ovary. Sex steroids largely exert their effects by binding to intracellular receptors, which subsequently interact with sex steroid-responsive elements in the

genome to modulate the transcription of target genes. Recent work, particularly in rodents and songbirds, has identified sex steroid-responsive neural circuits that underlie the divergent mating behaviors of males versus females, as well as the genes downstream of sex steroids that mediate these behaviors. However, the identity of the sex-specific direct transcriptional targets of sex steroids that mediate sex-typical mating behaviors remains elusive (*Yang and Shah, 2014*).

It is generally viewed that differences between the sexes in mating behavior are robust and essentially irreversible in mammals and birds. Several studies, however, have challenged this view and showed that they exhibit some degree of sexual lability in mating behavior (*Edwards and Burge, 1971*; *Södersten, 1972*; *Adkins-Regan, 2009*; *Balthazart et al., 2010*; *Ball et al., 2014*; *Kimchi et al., 2007*). Interestingly, sexual lability in mating behavior can be seen much more frequently and completely in teleost fish, where experimental manipulations that alter the sex steroid milieu effectively lead to the reversal of sex-typical mating behaviors, even in adulthood (*Munakata and Kobayashi, 2010*; *Paul-Prasanth et al., 2013*; *Göppert et al., 2016*; *Ghosal and Sorensen, 2016*). Furthermore, a large number of teleost species spontaneously undergo phenotypic sex reversal, involving both behavioral and anatomical changes, in response to social and physiological stimuli (*Liu et al., 2017*; *Capel, 2017*). Despite several studies, the neural mechanisms responsible for the reversal of sex-typical mating behaviors remain unknown.

In the teleost species medaka (*Oryzias latipes*), we previously found that estrogen and androgen receptors are expressed almost exclusively in females in two brain regions: the supracommissural/posterior nucleus of the ventral telencephalic area (Vs/Vp) and the magnocellular/gigantocellular portion of the magnocellular preoptic nucleus (PMm/PMg) (*Hiraki et al., 2012*) (*Figure 1A,B*). These regions are therefore likely to represent female-specific target sites for both estrogen and androgen in the teleost brain. This finding, together with classic lesion and stimulation studies suggesting that Vs/Vp and PMm/PMg are involved in mating behavior (*Demski et al., 1975*; *Kyle and Peter, 1982*; *Koyama et al., 1984*; *Satou et al., 1984*), led us to search for genes that are directly activated by estrogen and/or androgen in these brain regions and mediate female-typical mating behavior. As a result, we identified a candidate gene, *npb* (encoding neuropeptide B; NPB), whose expression in Vs/Vp and PMm/PMg is virtually confined to females and dependent on gonadal estrogen (*Hiraki et al., 2014*). NPB, together with its close relative, neuropeptide W (NPW), was originally identified as a ligand for the orphan receptors GPR7 and GPR8 (now designated NPBWR1 and NPBWR2) (*Fujii et al., 2002*; *Brezillon et al., 2003*; *Tanaka et al., 2003*). NPB and NPW have been implicated in a wide array of physiological processes, including food intake, energy homeostasis, inflammatory pain response, regulation of pituitary hormones, and social interaction (*Sakurai, 2013*; *Watanabe and Yamamoto, 2015*); to our knowledge, however, there is no information regarding their role in mating behavior.

In the present study, we investigated whether the sexually dimorphic expression of *npb* (renamed *npba* after the discovery of its paralog named *npbb*) in the medaka brain is directly elicited by estrogen, can be reversed by changes in the adult sex steroid milieu, and is relevant to female-typical mating behavior. Our results demonstrate that NPB acts directly downstream of estrogen in a female-specific but reversible manner to mediate female sexual receptivity.

## Results

### Sexually dimorphic *npba* expression is independent of sex chromosome complement but dependent on direct transcriptional activation by estrogen

First, we estimated the magnitude of chromosomal and hormonal influences on the sexually dimorphic expression of *npba*. We addressed whether the pattern of *npba* expression coincides with the chromosomal sex and/or gonadal sex by producing sex-reversed medaka and examining *npba* expression in their brains. Real-time PCR performed on the whole brain revealed that sex-reversed XY females exhibited the same high level of overall *npba* expression as wild-type XX females, whereas sex-reversed XX males showed a much lower level of overall *npba* expression (p<0.0001 versus XX and XY females), comparable to that in wild-type XY males (*Figure 1C*). Consistent with these results, in situ hybridization analysis revealed abundant *npba* expression in Vs/Vp and PMm/PMg of sex-reversed XY females, similar to wild-type XX females, whereas *npba* expression in these

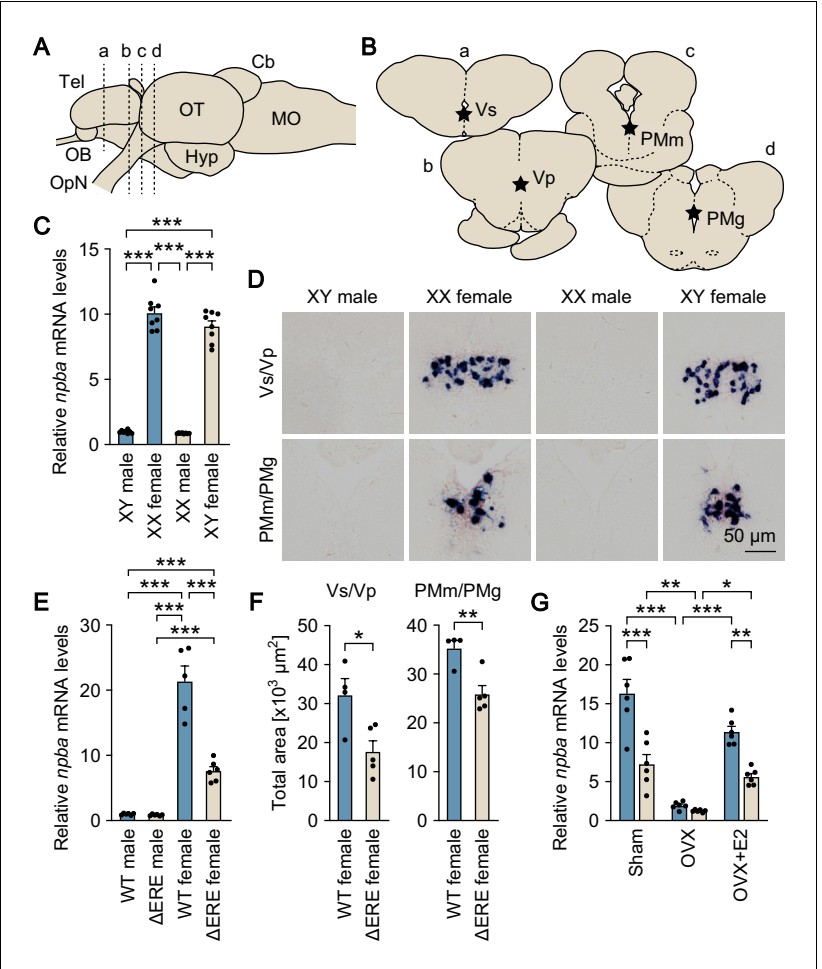

**Figure 1.** Sexually dimorphic *npba* expression is independent of sex chromosome complement but dependent on direct transcriptional activation by estrogen. (**A**) Lateral view (anterior to left) of the medaka brain showing approximate levels of sections in panel B. For abbreviations of brain regions, see *Supplementary file 1*. (**B**) Coronal sections showing the location of Vs/Vp and PMm/PMg. (**C**) Levels of *npba* expression in the whole brain of artificially sex-reversed XX males/XY females, and wild-type XY males/XX females as determined by real-time PCR (n = 8 per group). ***, p<0.001 (Bonferroni's *post hoc* test). (**D**) Representative micrographs of *npba* expression in Vs/Vp and PMm/PMg of sex-reversed XX males/XY females and wild-type XY males/XX females (n = 5 per group). Scale bars represent 50 μm. (**E**) Levels of *npba* expression in the whole brain of male and female mutants lacking the estrogen-responsive element in the *npba* promoter (ΔERE), and wild-type (WT) males and females as determined by real-time PCR (n = 6 per group, except WT females, where n = 5). ***, p<0.001 (Bonferroni's *post hoc* test). (**F**) Total area of *npba* expression in Vs/Vp and PMm/PMg of ΔERE (n = 5) and WT (n = 4) females. *, p<0.05; **, p<0.01 (unpaired *t*-test). (**G**) Levels of *npba* expression in the whole brain of ΔERE (beige columns) and WT (blue columns) females that were sham-operated (Sham), ovariectomized (OVX), or ovariectomized and treated with estradiol-17β (OVX +E2) as determined by real-time PCR (n = 6 per group). There were significant main effects of genotype ($F_{(1, 30)}$=45.03, p<0.0001) and treatment ($F_{(2, 30)}$=60.19, p<0.0001) and a significant interaction between genotype and treatment ($F_{(2, 30)}$=9.944, p=0.0005). *, p<0.05; **, p<0.01; ***, p<0.001 (Bonferroni's *post hoc* test). See also *Figure 1—figure supplement 1* and *Figure 1—figure supplement 2*.

DOI: https://doi.org/10.7554/eLife.39495.002

The following figure supplements are available for figure 1:

**Figure supplement 1.** Genetic scheme for the ΔERE mutant medaka.

DOI: https://doi.org/10.7554/eLife.39495.003

**Figure supplement 2.** Levels of estradiol-17β (E2) in the brains of ΔERE mutant and wild-type (WT) females.

DOI: https://doi.org/10.7554/eLife.39495.004

brain nuclei was not detected in either sex-reversed XX males or wild-type XY males (*Figure 1D*). Thus, the sexually dimorphic pattern of *npba* expression is independent of sex chromosome complement, but linked to gonadal phenotype.

Next, we investigated whether estrogen directly activates the transcription of *npba* in vivo by generating and analyzing ΔERE mutant medaka lacking the estrogen-responsive element (ERE) in the *npba* promoter, which has been shown to be functional in vitro (*Hiraki et al., 2014*) (*Figure 1—figure supplement 1*). Real-time PCR performed on the whole brain demonstrated that ΔERE mutant females had a significantly lower level of overall *npba* expression as compared with wild-type females (p<0.0001), whereas ΔERE mutant males had an even lower level (p<0.0001 versus wild-type and ΔERE mutant females), comparable to that in wild-type males (*Figure 1E*). In agreement with these results, in situ hybridization analysis revealed a smaller total area of *npba* expression in Vs/Vp and PMm/PMg of ΔERE mutant females as compared with wild-type females (p=0.0202 for Vs/Vp, and 0.0058 for PMm/PMg) (*Figure 1F*), whereas *npba* expression in these nuclei was not detected in ΔERE or wild-type males. We also evaluated *npba* expression in the whole brain of ΔERE mutant females that were sham-operated, ovariectomized, or ovariectomized and treated with estradiol-17β (E2; the primary estrogen in vertebrates including teleosts) by real-time PCR. In the mutant female brain, *npba* expression was significantly reduced by ovariectomy and restored by E2 treatment, but E2 was not able to restore *npba* expression to the level present in wild-type females (p=0.0023 for the mutant versus wild-type females) (*Figure 1G*). Taken together, these results demonstrate that estrogen elicits the female-specific expression of *npba* by direct transcriptional activation through the ERE present in the *npba* promoter, although other mechanisms are also likely involved in this estrogen effect.

The possibility that decreased *npba* expression in ΔERE mutant females was due to a decrease in estrogen levels in their brains was ruled out by the observation that ΔERE mutant females had brain levels of E2 comparable to those of wild-type females (p=0.782) (*Figure 1—figure supplement 2*).

## Sexually dimorphic *npba* expression can be reversed by altering the sex steroid milieu

We determined whether the sexual phenotype of *npba* expression can be reversed from the female to the male pattern, and vice versa, by altering the adult sex steroid milieu. In a previous study, we showed that *npba* expression in Vs/Vp and PMm/PMg of females was abolished by ovariectomy but then restored by estrogen replacement (*Hiraki et al., 2014*). Here, we first tested whether *npba* expression in these brain nuclei can be induced in males by castration and sex steroid supplementation. Indeed, in situ hybridization results showed that treating castrated males with E2 induced *npba* expression in both Vs/Vp and PMm/PMg (p=0.0013 for Vs/Vp and 0.0006 for PMm/PMg) (*Figure 2—figure supplement 1*).

The effects of altering the sex steroid milieu on the sexually dimorphic pattern of *npba* expression was further evaluated in fish with intact gonads. In situ hybridization results showed that treating ovary-intact adult females continuously with 11-ketotestosterone (KT; the most prominent, non-aromatizable androgen in teleosts) caused a gradual decline in *npba* expression in Vs/Vp, decreasing both the number of *npba*-expressing neurons (p<0.0001 on days 5 and 9) and the total area of *npba* expression (p=0.0002 on day 2; p<0.0001 on days 5 and 9) to undetectable or nearly undetectable levels over the treatment period (*Figure 2A–C*). A significant decline in the area of *npba* expression was also observed in PMm/PMg (p=0.0068 for the total area on day 9), although the number of *npba*-expressing neurons was unchanged (*Figure 2A–C*). Treating ovary-intact adult females with an aromatase inhibitor (AI; aromatase converts androgen to estrogen) yielded essentially the same results (p<0.0001 for number of neurons in Vs/Vp on days 2, 5, and 9; p=0.0063 and 0.0025 for total area in Vs/Vp on days 5 and 9, respectively; p=0.0108 for total area in PMm/PMg on day 9) (*Figure 2D–F*). To aid in interpreting these results, we measured brain levels of E2 in females treated with KT in exactly the same way as for the *npba* expression analysis. The results showed that E2 levels fell to less than 5% of untreated controls within 2 days and remained low during the remainder of the treatment period (p<0.0001 on days 2, 5, and 9), demonstrating a substantial decrease in E2 levels in the brain of KT-treated females (*Figure 2—figure supplement 2*).

In contrast, treating testis-intact adult males continuously with E2 resulted in the induction of *npba* expression in both Vs/Vp and PMm/PMg, where both the number of *npba*-expressing neurons and the total area of *npba* expression steadily increased over time during treatment (p<0.0001 for

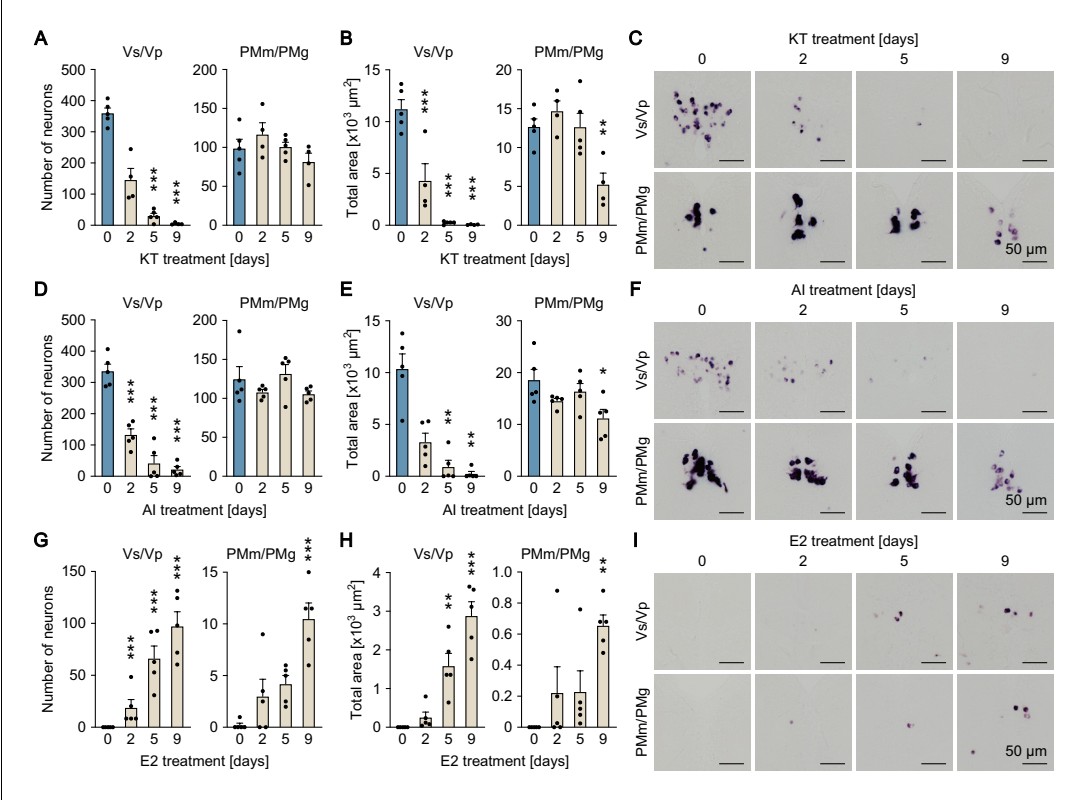

**Figure 2.** Sexually dimorphic *npba* expression can be reversed by altering the sex steroid milieu. Temporal changes in *npba* expression in Vs/Vp and PMm/PMg of 11-ketotestosterone (KT)-treated females (**A, B, C**), aromatase inhibitor (AI)-treated females (**D, E, F**), and estradiol-17β (**E2**)-treated males (**G, H, I**) (n = 5 per treatment and sampling day). (**A, D, G**) Number of *npba*-expressing neurons in Vs/Vp and PMm/PMg. ***, p<0.001 (versus day 0, Dunnett's *post hoc* test). (**B, E, H**) Total area of *npba* expression in Vs/Vp and PMm/PMg. *, p<0.05; **, p<0.01; ***, p<0.001 (versus day 0, Dunnett's *post hoc* test). (**C, F, I**) Representative micrographs showing *npba* expression in Vs/Vp and PMm/PMg. Scale bars represent 50 μm. See also *Figure 2— figure supplement 1* and *Figure 2—figure supplement 2*.

DOI: https://doi.org/10.7554/eLife.39495.005

The following figure supplements are available for figure 2:

**Figure supplement 1.** Effects of castration and sex steroid supplementation on *npba* expression in Vs/Vp and PMm/PMg of males.
DOI: https://doi.org/10.7554/eLife.39495.006

**Figure supplement 2.** Levels of estradiol-17β (E2) in the brains of females treated with 11-ketotestosterone (KT).
DOI: https://doi.org/10.7554/eLife.39495.007

number of neurons in Vs/Vp on days 2, 5, and 9; p=0.0013 and<0.0001 for total area in Vs/Vp on days 5 and 9, respectively; p<0.0001 for number of neurons in PMm/PMg on day 9; p=0.0023 for total area in PMm/PMg on day 9) (*Figure 2G–I*). Thus, the sexually dimorphic pattern of *npba* expression can be reversed in both directions in response to changes in the sex steroid milieu.

## Female-specific Npba likely acts widely in the brain and spinal cord

Next, we investigated the sites of action of the Npba peptide produced specifically by females. First, the axonal projections of *npba*-expressing neurons were determined by immunohistochemical detection of Npba and compared between the sexes. The specificity of the anti-Npba antibody was ascertained by a double-labeling experiment with immunohistochemistry and in situ hybridization, where it selectively identified neurons expressing the *npba* transcript (*Figure 3—figure supplement 1*). In addition, this antibody was found not to cross-react with Npbb as confirmed by the complete absence of labeling with this antibody in *npba* knockout females (*Figure 3—figure supplement 1*). Although the focus of the present study is female-specific subsets of *npba*-expressing neurons in Vs/ Vp and PMm/PMg and these neurons account for a large proportion of overall *npba* expression in the female brain, *npba* is expressed in both sexes in several brain nuclei, including Pbl in the

ventrolateral preoptic area (*Hiraki et al., 2014*). Consistent with this fact, Npba-immunoreactive axons arising from the neuronal cell bodies in Pbl and extending through the hypothalamus into the pituitary were observed in both sexes (*Figure 3A*). In females, additional Npba-immunoreactive axons were found in many areas of the central nervous system, with particular abundance in the thalamus, hypothalamus, optic tectum, midbrain tegmentum, medulla oblongata, and spinal cord (*Figure 3A*).

Axonal projections were additionally evaluated by generating and examining transgenic medaka in which *npba*-expressing neurons were labeled with GFP (*npba*-GFP transgenic medaka). The specificity of GFP expression was confirmed by double in situ hybridization detecting GFP and endogenous *npba* transcripts, which demonstrated that the distribution of the two transcripts almost completely overlapped (*Figure 3—figure supplement 2*). Fluorescence imaging of the transgenic fish yielded results that were fairly consistent with immunohistochemistry: that is, a subset of axons common to both sexes and a subset specific to females were apparent (*Figure 3B*). In addition, imaging of whole brains and spinal cords revealed that the female-specific axons reached the posterior end of the spinal cord and that all female-specific axons most probably originated from the female-specific *npba*-expressing neurons in Vs/Vp and PMm/PMg.

However, we could not rule out the possibility that additional female-specific *npba*-expressing neurons were present in the medulla oblongata and/or spinal cord (regions that have not previously been studied) and were the source of the female-specific axons therein. Accordingly, we surveyed *npba*-expressing neurons in these regions by in situ hybridization. Although a large number of *npba*-expressing neurons were widely scattered over the medulla oblongata and spinal cord, all of these subsets of neurons were common to both sexes and were not confined to, or predominant in, females (*Figure 3—figure supplement 3*), thereby showing that the female-specific axons did not originate from these regions.

The sites of action of Npba were further delineated by identifying the gene encoding the NPB/ NPW receptor (NPBWR) in medaka and assessing the spatial distribution of its expression. Screening of an expressed sequence tag (EST) library generated from medaka brain, followed by 5′-rapid amplification of cDNA ends (5′-RACE), led to the isolation of a full-length cDNA of 4045 bp (deposited in DDBJ/EMBL/GenBank with accession number LC375958), whose deduced amino acid sequence shared a high degree of identity with NPBWR2 in other species (*Figure 3—figure supplement 4*). Phylogenetic tree analysis demonstrated that this cDNA encoded a medaka protein orthologous to NPBWR2 and paralogous to NPBWR1 of other species (*Figure 3C*). No other *NPBWR1/ NPBWR2*-like genes were found in our EST library or public medaka genome/EST databases, indicating the absence of NPBWR1 in medaka as in zebrafish (*Danio rerio*) (*Bu et al., 2016*).

In situ hybridization identified *npbwr2* expression in the following nuclei in both males and females: Vv, Vl, and Vs/Vp in the ventral telencephalon; Dm/Dl in the dorsal telencephalon; PMp, PPa, and PPp in the preoptic area, rHd and lHd in the habenula; VM and DP/CP in the thalamus; PGZ3 in the optic tectum; NDLI, NAT, NVT, NRL, NPT, and NRP in the hypothalamus; IR/MR/IQ and DT/TS/is in the midbrain tegmentum; ra and gc/RS in the brain stem; the peripheral region of the anterior pituitary; and the medial part of the dorsal horn and the lateral part of the ventral horn throughout the spinal cord (*Figure 3D*; abbreviations of brain and spinal cord regions and brain nuclei are defined in *Supplementary file 1*). There were no overt differences in the distribution pattern of *npbwr2* expression between the sexes.

Collectively, these results suggest that Npba that is produced female-specifically in Vs/Vp and PMm/PMg is transported to and acts in many different brain areas and the spinal cord.

## Medaka possess an additional NPB gene, designated *npbb*

Via a survey of the medaka genome and EST databases, we identified a previously uncharacterized gene encoding a protein highly homologous to, but distinct from, Npba. Structural analysis of the protein sequence predicted a mature NPB polypeptide at residues 26–54, which shared a high degree of sequence identity with known mature NPB polypeptides in medaka and other species (*Figure 4—figure supplement 1*). The protein sequence was also predicted to contain a signal peptide at the N-terminus with a cleavage site between residues 25 and 26 (*Figure 4—figure supplement 1*). Phylogenetic tree analysis indicated that this protein represents an additional NPB and was thus designated Npbb with re-designation of the original Npb as Npba (*Figure 4A*).

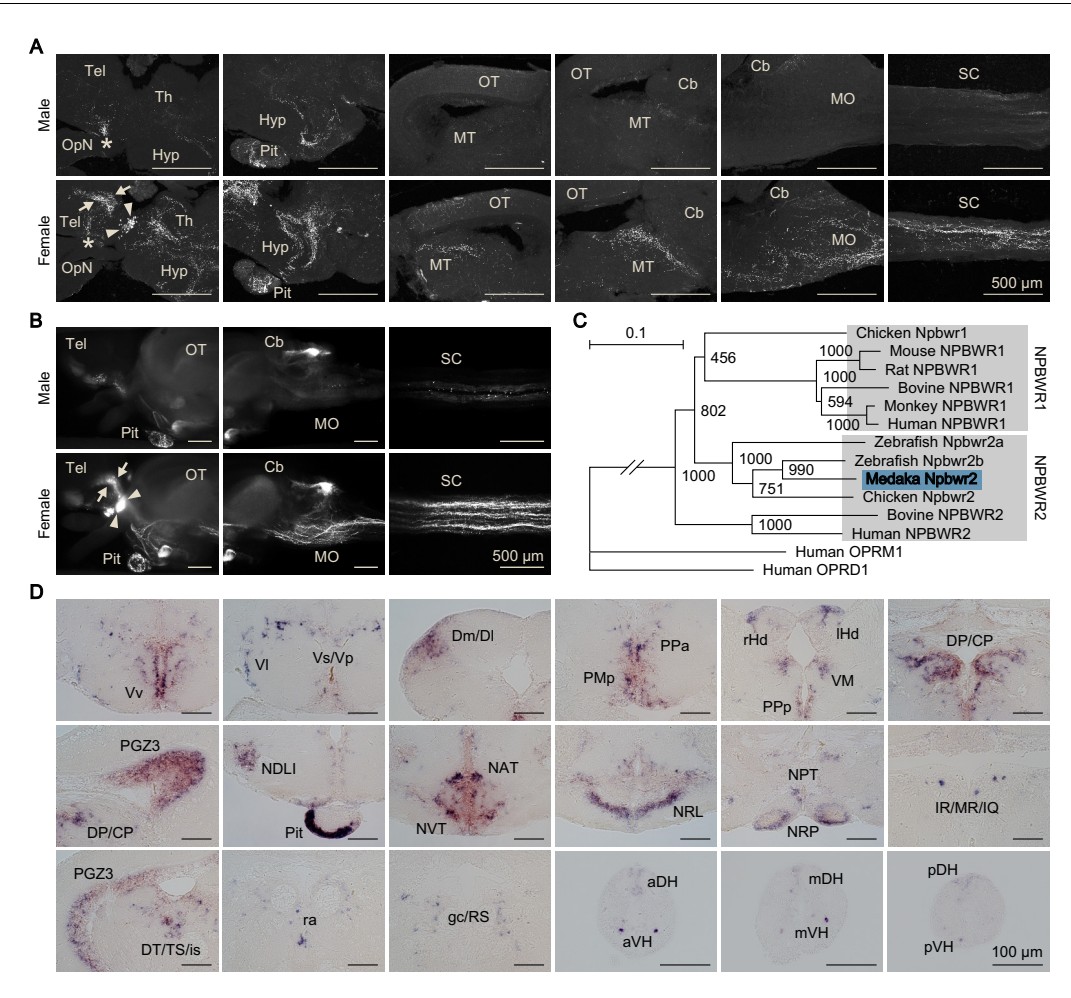

**Figure 3.** Female-specific Npba likely acts widely in the brain and spinal cord. (**A**) Comparison of the distribution of Npba-immunoreactive axons in the brain, pituitary, and spinal cord between males and females. All images are sagittal sections with anterior to the left. Arrows and arrowheads indicate female-specific Npba-immunoreactive neuronal cell bodies in Vs/Vp and PMm/PMg, respectively. Asterisks indicate Npba-immunoreactive neuronal cell bodies in Pbl occurring in both sexes. Scale bars represent 500 μm. For abbreviations of brain regions, see *Supplementary file 1*. (**B**) Comparison of the distribution of GFP-labeled axons in the brain, pituitary, and spinal cord between *npba*-GFP transgenic males and females. Images in the left and middle panels are lateral views with anterior to the left; images in the right panels are horizontal views with anterior to the left. Arrows and arrowheads indicate female-specific GFP-labeled neuronal cell bodies in Vs/Vp and PMm/PMg, respectively. Scale bars represent 500 μm. (**C**) Phylogenetic tree of NPBWR1 and NPBWR2. The number at each node indicates bootstrap values for 1000 replicates. Human opioid receptors μ1 (OPRM1) and δ1 (OPRD1) were used as the outgroup for tree reconstruction. Scale bar represents 0.1 substitution per site. For species names and GenBank accession numbers, see *Supplementary file 3*. (**D**) Distribution of *npbwr2* expression in the brain, pituitary, and spinal cord. All images are coronal sections. Images of only males are presented, because there were no obvious sex differences in the distribution of expression (n = 5 per sex). Scale bars represent 100 μm. For abbreviations of brain and spinal cord regions and brain nuclei, see *Supplementary file 1*. See also *Figure 3—figure supplement 1*, *Figure 3—figure supplement 2*, *Figure 3—figure supplement 3*, and *Figure 3—figure supplement 4*.

DOI: https://doi.org/10.7554/eLife.39495.008

The following figure supplements are available for figure 3:

**Figure supplement 1.** Verification of the specificity of the anti-Npba antibody.

DOI: https://doi.org/10.7554/eLife.39495.009

**Figure supplement 2.** Generation and verification of *npba*-GFP transgenic medaka.

DOI: https://doi.org/10.7554/eLife.39495.010

**Figure supplement 3.** Distribution of *npba*-expressing neurons in the medaka medulla oblongata and spinal cord.

DOI: https://doi.org/10.7554/eLife.39495.011

**Figure supplement 4.** Sequence information for medaka *npbwr2*.

DOI: https://doi.org/10.7554/eLife.39495.012

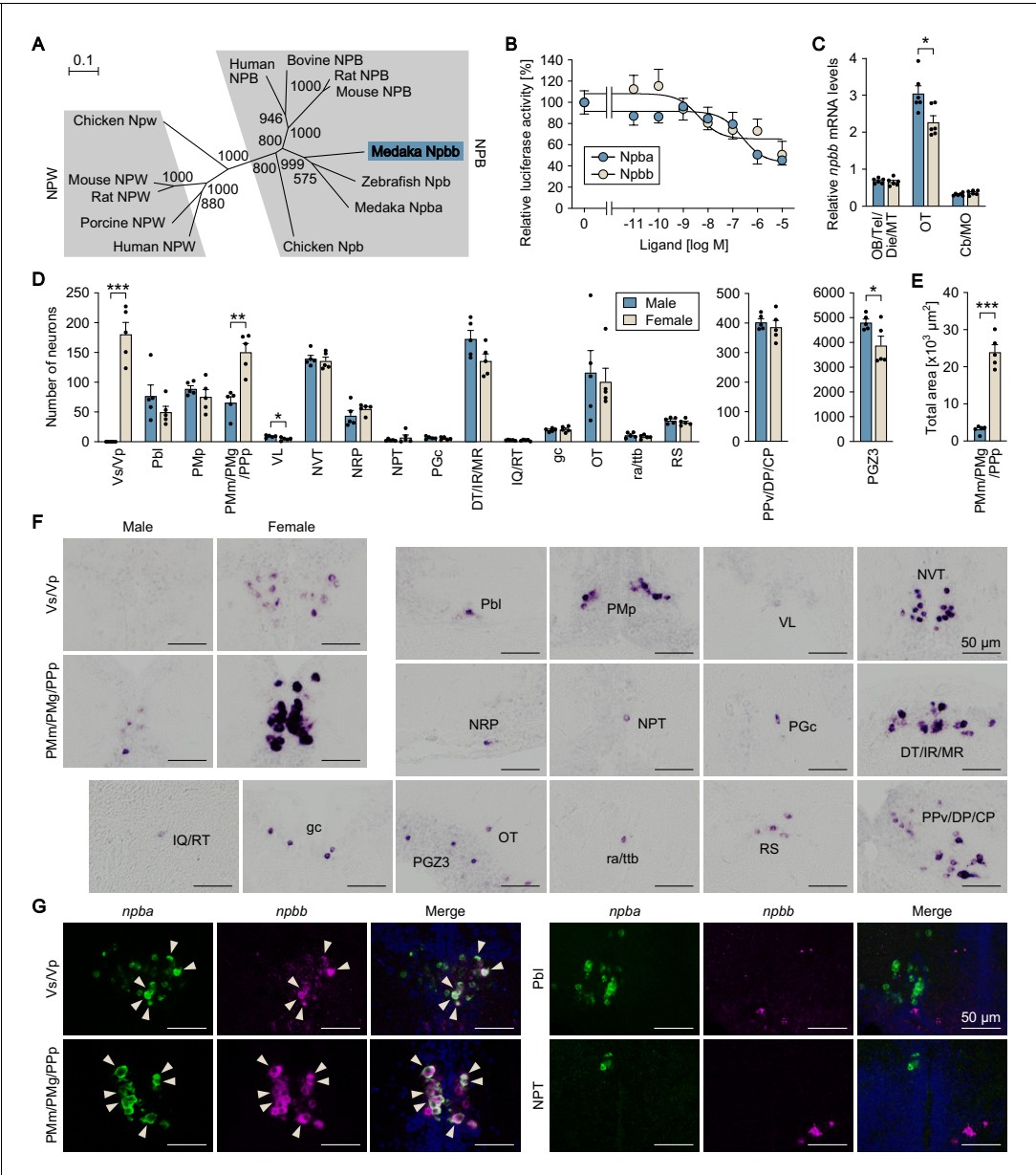

**Figure 4.** Medaka possess an additional NPB gene, designated *npbb*. (**A**) Phylogenetic tree of NPB and NPW. The number at each node indicates bootstrap values for 1000 replicates. Scale bar represents 0.1 substitution per site. For species names and GenBank accession numbers, see *Supplementary file 3*. (**B**) Ability of medaka Npba and Npbb to activate medaka Npbwr2. Receptor activation was assessed by measuring cAMP-responsive element-driven luciferase activity in cells transfected with Npbwr2. The *x*-axis shows the concentration of Npba and Npbb, and the *y*-axis shows luciferase activity as a percentage of that observed in the absence of Npba/Npbb. (**C**) Overall levels of *npbb* expression in the male (blue columns) and female (beige columns) brain dissected into three parts as determined by real-time PCR (n = 6 per sex). *, p<0.05 (unpaired *t*-test). For abbreviations of brain regions, see *Supplementary file 1*. (**D**) Distribution of *npbb*-expressing neurons in the male (blue columns) and female (beige columns) brain (n = 5 per sex). The data are split into three graphs for clarity. *, p<0.05; **, p<0.01; ***, p<0.001 (unpaired *t*-test). For abbreviations of brain nuclei, see *Supplementary file 1*. (**E**) Total area of *npbb* expression in PMm/PMg/PPp in males (blue column) and females (beige column) (n = 5 per sex). ***, p<0.001 (unpaired *t*-test). (**F**) Representative micrographs showing *npbb*-expressing neurons in different brain nuclei. Micrographs of both sexes are shown for Vs/Vp and PMm/PMg/PPp, where *npbb* expression is confined or almost confined to females. Micrographs of males only are shown for other nuclei, where sex differences were not detected or, if present, were not sufficiently demonstrated by photographs. (**G**) Coexpression of *npba* and *npbb* in the same neurons in Vs/Vp and PMm/PMg, but not in other brain nuclei. In each set of panels, the left and middle ones show images of *npba* (green) and *npbb* (magenta) expression, respectively, in the same sections; the right ones show the merged images with nuclear counterstaining (blue). Arrowheads indicate representative neurons coexpressing *npba* and *npbb*. Scale bars represent 50 μm. For abbreviations of brain nuclei, see *Supplementary file 1*. See also *Figure 4—figure supplement 1* and *Figure 4—figure supplement 2*.

DOI: https://doi.org/10.7554/eLife.39495.013

*Figure 4 continued on next page*

*Figure 4 continued*

The following figure supplements are available for figure 4:

**Figure supplement 1.** Sequence information for medaka *npbb*.

DOI: https://doi.org/10.7554/eLife.39495.014

**Figure supplement 2.** Comparison of syntenic relationships of genes in the vicinity of gar, arowana, catfish, and medaka NPB genes.

DOI: https://doi.org/10.7554/eLife.39495.015

Database searches using the medaka NPB genes as queries led to the identification of two distinct NPB genes in many other teleost species, including arowana (*Scleropages formosus*), catfish (*Ictalurus punctatus*), salmon (*Salmo salar*), trout (*Oncorhynchus mykiss*), pufferfish (*Tetraodon nigroviridis*), and tilapia (*Oreochromis niloticus*), while only one NPB gene was identified in non-teleost species, including gar (*Lepisosteus oculatus*), a non-teleost ray-finned fish. Comparative synteny analysis of NPB genes in the gar, basal (early-branching) teleost (arowana and catfish), and medaka genomes revealed that, in the basal teleost and medaka genomes, *npba* is located in a segment of conserved synteny containing *sstr2*, *sox9*, *rnf213*, *pcyt2*, and *mafg*, while *npbb* is in a segment of conserved synteny containing *slc26a11*, *sgsh*, *gnav1*, and *pcyt2*. On the other hand, the NPB gene in the gar genome is located in a segment containing all of these genes (*Figure 4—figure supplement 2*). These lines of evidence strongly suggest that the two NPB genes arose from the whole-genome duplication that occurred early in teleost evolution (*Amores et al., 1998*). No other *NPB/NPW*-like genes were found in our EST library or public medaka genome/EST databases, indicating the absence of NPW in medaka as in zebrafish (*Bu et al., 2016*). Receptor activation assays revealed that both Npba and Npbb dose-dependently activated medaka Npbwr2, resulting in a decrease in intracellular cAMP levels, as reported for NPBWR1/2 of other species (*Fujii et al., 2002*; *Tanaka et al., 2003*). Npbb displayed greater potency than Npba with a 50% inhibitory concentration (IC50) of 4.3 nM versus 240 nM (*Figure 4B*).

Next, we examined the spatial expression of *npbb* with a focus on possible differences between the sexes and colocalization with *npba* expression. First, the relative levels of regional *npbb* expression were measured by real-time PCR and compared between male and female brains dissected into three parts: (i) the olfactory bulb, telencephalon, diencephalon, and midbrain tegmentum; (ii) the optic tectum; and (iii) the cerebellum and medulla oblongata.Expression of *npbb* was observed in all three parts, but was highest in the optic tectum, where males had slightly higher levels than females (p=0.014) (*Figure 4C*). Subsequent examination by in situ hybridization revealed that *npbb* was expressed in the following nuclei: Vs/Vp in the ventral telencephalon; Pbl, PMp, and PMm/PMg/PPp in the preoptic area (note that *npbb* was continuously expressed across PMm/PMg and PPp, whereas *npba* was expressed separately in PMm/PMg and PPp); VL and PPv/DP/CP in the pretectum/thalamus; NVT, NRP, NPT, and PGc in the hypothalamus; DT/IR/MR and IQ/RT in the midbrain tegmentum; gc, ra/ttb, and RS in the brain stem; and OT and PGZ3 in the optic tectum (*Figure 4D, F*). A marked difference between the sexes was observed in Vs/Vp (p<0.0001), where *npbb* expression was observed in females, but completely absent in males (*Figure 4D,F*). Similarly, *npbb*-expressing neurons in PMm/PMg/PPp were more abundant in number and displayed a much larger total area of expression in females than in males (p=0.0011 for number of neurons; p<0.0001 for total area) (*Figure 4D–F*). In contrast, in VL and PGZ3, a marginally higher number of *npb*-expressing neurons were seen in males than in females (p=0.020 for VL, and 0.045 for PGZ3) (*Figure 4D*).

The simultaneous detection of *npba* and *npbb* expression showed that these genes are expressed in the same subset of neurons within Vs/Vp and PMm/PMg/PPp; in contrast, no neurons in other brain nuclei were found to express both genes (*Figure 4G*). Taken together, these findings show that there is an additional NPB gene, *npbb*, in medaka that is coexpressed with *npba* in a female-biased manner in Vs/Vp and PMm/PMg/PPp.

## Sexually dimorphic *npbb* expression can also be reversed by altering the sex steroid milieu

Following identification of *npbb*, we investigated whether the sexually dimorphic pattern of *npbb* expression in Vs/Vp and PMm/PMg/PPp can be reversed by altering the adult sex steroid milieu, as observed for *npba*. First, we evaluated the effects of sex steroids on *npbb* expression by means of

gonadectomy and sex steroid supplementation, followed by in situ hybridization analysis. In adult females, ovariectomy caused a significant reduction in *npbb* expression in both Vs/Vp (p=0.0271) and PMm/PMg/PPp (p=0.0041), which was restored by treatment with E2 (p=0.0091 for Vs/Vp, and 0.0173 for PMm/PMg) but not by KT (*Figure 5—figure supplement 1*). In adult males, E2 treatment following castration induced a few *npbb*-expressing neurons in Vs/Vp (p<0.0001), whereas *npbb* expression in PMm/PMg/PPp did not show clear responses to any treatment (*Figure 5—figure supplement 1*).

The effects of the altered sex steroid milieu on *npbb* expression was further tested in fish with intact gonads. Both KT and AI treatment of ovary-intact adult females resulted in a gradual decline in *npbb* expression in Vs/Vp, both in the number of *npbb*-expressing neurons and in the total area of *npbb* expression, to undetectable or nearly undetectable levels (p<0.0001 for number of neurons in KT-treated fish on days 5 and 9; p=0.0235,<0.0001, and <0.0001 for total area in KT-treated fish on days 2, 5, and 9, respectively; p=0.0002 and<0.0001 for number of neurons in AI-treated fish on days 5 and 9, respectively; p=0.0013 and 0.0009 for total area in AI-treated fish on days 5 and 9, respectively) (*Figure 5A–F*). KT treatment, but not AI treatment, also led to a significant decrease in *npbb* expression in PMm/PMg/PPp (p=0.0038 for number of neurons in KT-treated fish on day 9; p=0.0021 and<0.0001 for total area in KT-treated fish on days 5 and 9, respectively) (*Figure 5A–F*). E2 treatment of testis-intact adult males caused no clear changes in *npbb* expression in either Vs/Vp or PMm/PMg/PPp, but a few *npbb*-expressing neurons were newly induced in Vs/Vp (p=0.0369) (*Figure 5G–I*). Collectively, these results show that, similar to *npba* expression, the sexually dimorphic pattern of *npbb* expression can be reversed, at least in part, in response to changes in the sex steroid milieu.

Search for putative EREs in the *npbb* promoter led us to identify a putative ERE at a similar position as in the *npba* promoter. In addition, the *npb* promoter in gar and the *npba*/*npbb* promoters in several teleost species examined were also found to contain a putative ERE at a similar position (*Figure 5—figure supplement 2*).

## Npba/Npbb/Npbwr2 signaling is involved in female sexual receptivity

To examine the role of *npba* and *npbb* in female mating behavior, we generated *npba* knockout (*npba*-/-) and *npbb* knockout (*npbb*-/-) medaka by transcription activator-like effector nuclease (TALEN)-mediated genome editing (*Figure 6—figure supplement 1*) and observed the mating behavior of females. Most homozygous females of both knockout strains spawned successfully, similar to wild-type and heterozygous females. However, subsequent quantitative behavioral testing uncovered a latent abnormality in *npba* knockout strain. The mating behavior of medaka consists of a sequence of stereotyped actions that are easily quantified (*Ono and Uematsu, 1957*; *Walter and Hamilton, 1970*). The sequence begins with the male approaching and following the female closely. The male then performs a courtship display, in which he swims quickly in a circular pattern in front of the female. If she is receptive, the male grasps her with his dorsal and anal fins (termed 'wrapping'), and they quiver together (termed 'quivering') until eggs and sperm are released. If the female is not receptive, she either assumes a rejection posture in which she raises her head or rapidly moves away from the male. Here, we found a significant increase in latencies from the first courtship display to the first wrapping and to the wrapping that resulted in spawning for *npba*-/- females as compared with *npba*+/+ and *npba*+/- females (p=0.0256 and 0.0002, respectively, for the latency to the first wrapping; p=0.0104 and 0.0026, respectively, for the latency to the wrapping that resulted in spawning) (*Figure 6A*). In contrast, *npbb*-/- females compared with controls showed no significant difference in any of the behavioral parameters (*Figure 6B*). To further evaluate the role of NPB signaling, we generated *npba*/*npbb* double knockout (*npba*-/-/*npbb*-/-) and *npbwr2* knockout (*npbwr2*-/-) medaka (*Figure 6—figure supplement 1*) and assessed the mating behavior of these females. Homozygous females, as well as males, of both strains were found to be fertile, sexually active, and to spawn successfully. However, as was the case with *npba*-/- females, a significant increase in the latency from the first courtship display to the first wrapping was observed for both *npba*-/-/*npbb*-/- and *npbwr2*-/- females as compared with wild-type females (p=0.0064 for *npba*-/-/*npbb*-/-, and 0.0002 for *npbwr2*-/-) (*Figure 7A*). *npbwr2*-/- females also showed a significant increase in latency from the first courtship display to the wrapping that resulted in spawning (p=0.0013) (*Figure 7A*). Although not statistically significant, *npba*-/-/*npbb*-/- females showed a similar trend (p=0.0609) (*Figure 7A*). In addition, there was a significant increase in the proportion of females that

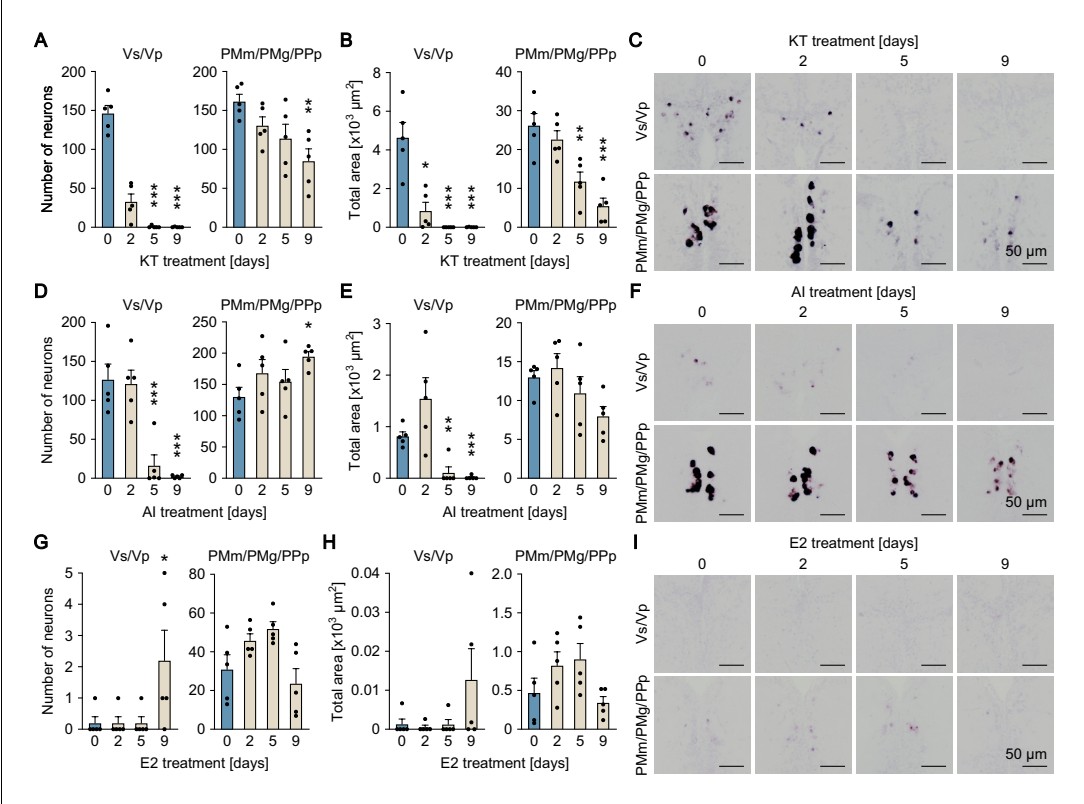

**Figure 5.** Sexually dimorphic *npbb* expression can also be reversed by altering the sex steroid milieu. Temporal changes in *npbb* expression in Vs/Vp and PMm/PMg of 11-ketotestosterone (KT)-treated females (A, B, C), aromatase inhibitor (AI)-treated females (D, E, F), and estradiol-17β (E2)-treated males (G, H, I) (n = 5 per treatment and sampling day). (A, D, G) Number of *npbb*-expressing neurons in Vs/Vp and PMm/PMg. *, $p<0.05$; **, $p<0.01$; ***, $p<0.001$ (versus day 0, Dunnett's *post hoc* test). (B, E, H) Total area of *npbb* expression in Vs/Vp and PMm/PMg. *, $p<0.05$; **, $p<0.01$; ***, $p<0.001$ (versus day 0, Dunnett's *post hoc* test). (C, F, I) Representative micrographs showing *npbb* expression in Vs/Vp and PMm/PMg. Scale bars represent 50 μm. See also *Figure 5—figure supplement 1* and *Figure 5—figure supplement 2*.

DOI: https://doi.org/10.7554/eLife.39495.016

The following figure supplements are available for figure 5:

**Figure supplement 1.** Effects of gonadectomy and sex steroid supplementation on *npbb* expression in Vs/Vp and PMm/PMg.

DOI: https://doi.org/10.7554/eLife.39495.017

**Figure supplement 2.** Putative estrogen-responsive elements (EREs) predicted in the proximal promoter regions of gar, arowana, catfish, medaka, and tilapia NPB genes.

DOI: https://doi.org/10.7554/eLife.39495.018

spawned without any preceding courtship display from the male for both *npba*[-/-]/*npbb*[-/-] (p=0.0405) and *npbwr2*[-/-] (p=0.0010) strains (*Figure 7A*).

Behavioral time-series data sets were further analyzed using Kaplan-Meier plots with the inclusion of fish that did not spawn within the test period. Although the significant difference disappeared in the latency from the first courtship display to the wrapping that resulted in spawning between *npba*[+/-] and *npba*[-/-], all other significant differences detected were also detected by this analysis, supporting the robustness of our results (the latency to the first wrapping; p=0.0480 for *npba*[+/+] versus *npba*[-/-] females, p=0.0291 for *npba*[+/-] versus *npba*[-/-] females, p=0.0162 for wild-type versus *npba*[-/-]/*npbb*[-/-] females, p<0.0002 for wild-type versus *npbwr2*[-/-] females: the latency to the wrapping that resulted in spawning; p=0.0129 for *npba*[+/+] versus *npba*[-/-] females, p=0.1158 for *npba*[+/-] versus *npba*[-/-] females, p=0.1068 for wild-type versus *npba*[-/-]/*npbb*[-/-] females, p=0.0006 for wild-type versus *npbwr2*[-/-] females) (*Figure 6C,D* and *Figure 7B*). Collectively, these results indicate that NPB signaling affects female receptivity to male courtship.

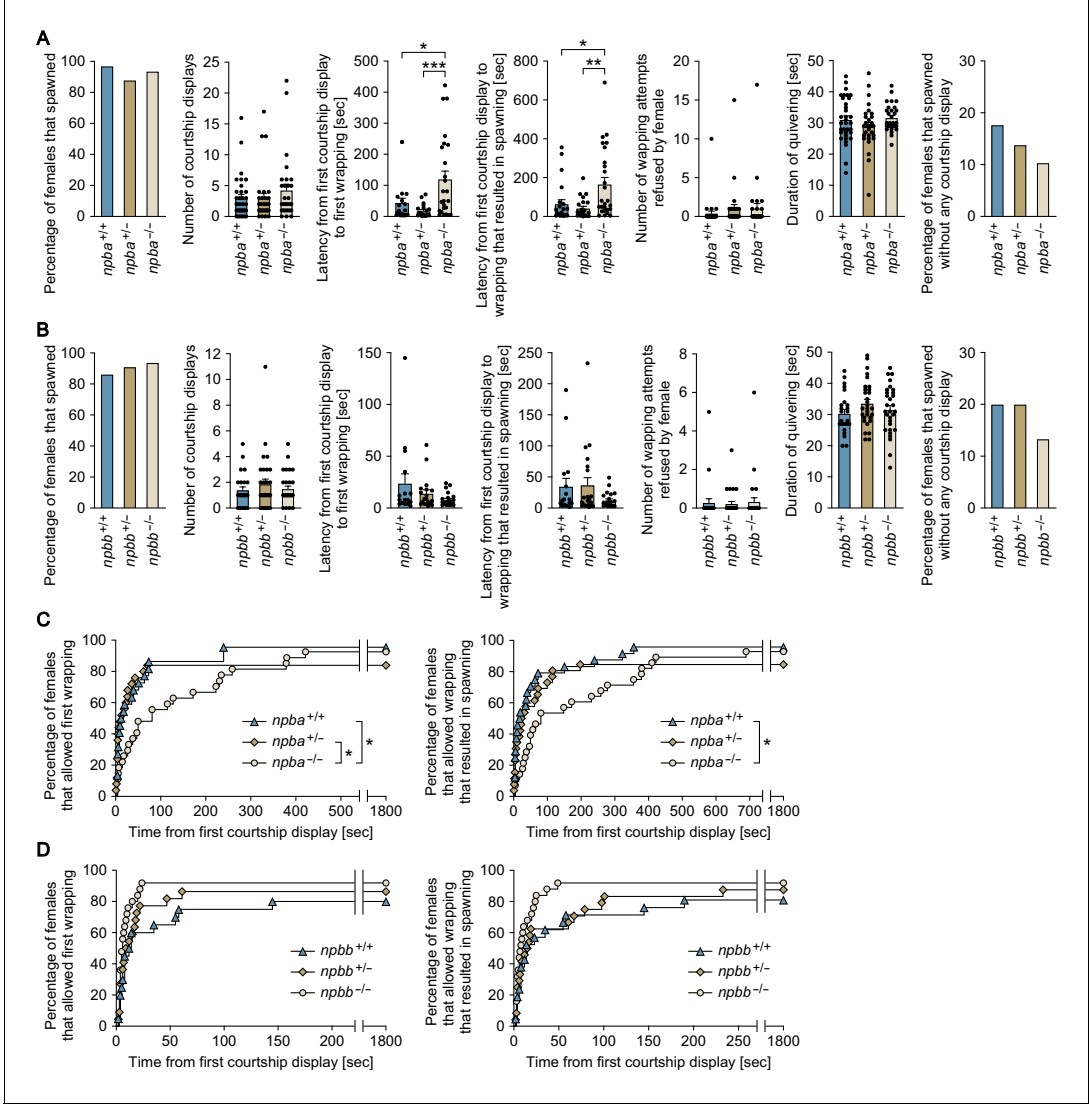

**Figure 6.** *npba* is involved in female sexual receptivity. (**A, B**) Various parameters in the mating behavior of *npba* (**A**) and *npbb* (**B**) single knockout females were measured and compared with wild-type females (n = 34, 33, and 31 for *npba*⁺/⁺, *npba*⁺/⁻, and *npba*⁻/⁻ females, respectively; n = 29, 33, and 32 for *npbb*⁺/⁺, *npbb*⁺/⁻, and *npbb*⁻/⁻ females, respectively). Blue, ocher, and beige columns represent wild-type, heterozygous knockout, and homozygous knockout females, respectively. *, p<0.05; **, p<0.01; ***, p<0.001 (Bonferroni's *post hoc* test). (**C, D**) The latency data for *npba* (**C**) and *npbb* (**D**) single knockouts were further analyzed using Kaplan-Meier plots. Blue triangles, ocher diamonds, and beige circles represent wild-type, heterozygous knockout, and homozygous knockout females, respectively. *, p<0.05 (Gehan-Breslow-Wilcoxon test with Bonferroni's correction). See also *Figure 6—figure supplement 1*.

DOI: https://doi.org/10.7554/eLife.39495.019

The following figure supplement is available for figure 6:

**Figure supplement 1.** Genetic scheme for the *npba*⁻/⁻, *npbb*⁻/⁻, and *npbwr2*⁻/⁻ mutant medaka.
DOI: https://doi.org/10.7554/eLife.39495.020

## Discussion

In a previous study, we showed that *npba* expression in Vs/Vp and PMm/PMg, two brain nuclei implicated in mating behavior, is specific to females in medaka (*Hiraki et al., 2014*). That finding, together with the fact that sex differences in mating behavior can be reversed in teleosts, led us to investigate whether *npba* has a role in female-typical mating behavior and whether its sex-specific expression pattern may be reversed between males and females. The present results indicate that

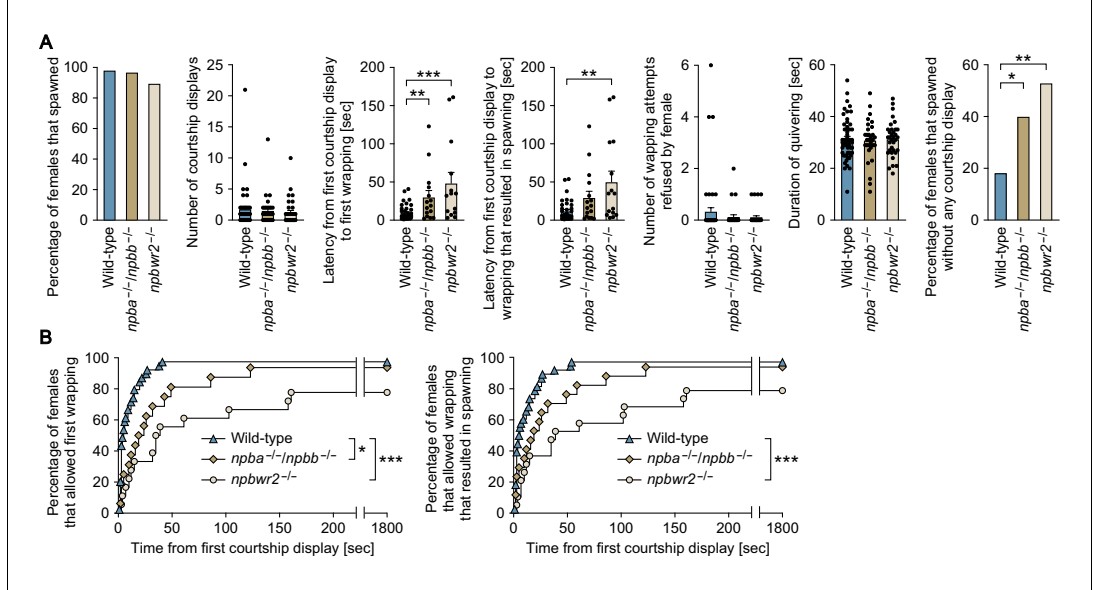

**Figure 7.** Npba/Npbb/Npbwr2 signaling is involved in female sexual receptivity. (**A**) Various parameters in the mating behavior of *npba/npbb* double knockout (*npba*[-/-]/*npbb*[-/-]) females (n = 31) and *npbwr2* knockout (*npbwr2*[-/-]) females (n = 38) were measured and compared with wild-type females (n = 55). Blue, ocher, and beige columns represent wild-type, *npba/npbb* double knockout, and *npbwr2* knockout females, respectively. \*, p<0.05; \*\*, p<0.01; \*\*\*, p<0.001 (Dunnett's *post hoc* test for data on number, latency, and duration; Fisher's exact test for data on percentage). (**B**) The latency data were further analyzed using Kaplan-Meier plots. Blue triangles, ocher diamonds, and beige circles represent wild-type, *npba/npbb* double knockout, and *npbwr2* knockout females, respectively. \*, p<0.05; \*\*\*, p<0.001 (Gehan-Breslow-Wilcoxon test with Bonferroni's correction). See *Video 1*, *Video 2*, and *Video 3*. Supplementary Data List.

DOI: https://doi.org/10.7554/eLife.39495.021

NPB signaling constitutes a female-specific but reversible component of the female sexual receptivity pathway.

Our previous study showed that female-specific *npba* expression is dependent on gonadal estrogen (*Hiraki et al., 2014*). In mammals and birds, many sex differences in behavior and gene expression in the brain are robustly influenced by sex chromosome complement, with chromosomal sex occasionally exerting an influence as large as that of gonadal sex steroids (*McCarthy and Arnold, 2011*; *Forger et al., 2016*). It therefore seemed possible that sex chromosome complement as well

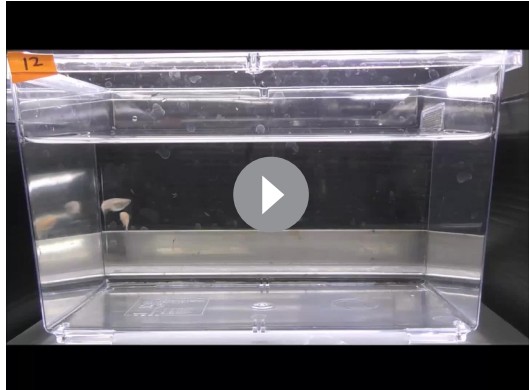

**Video 1.** A representative video showing the mating behavior of wild-type females.
DOI: https://doi.org/10.7554/eLife.39495.022

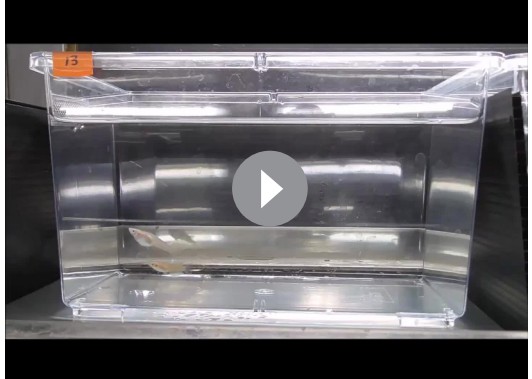

**Video 2.** A representative video showing the mating behavior of *npba/npbb* double knockout females.
DOI: https://doi.org/10.7554/eLife.39495.023

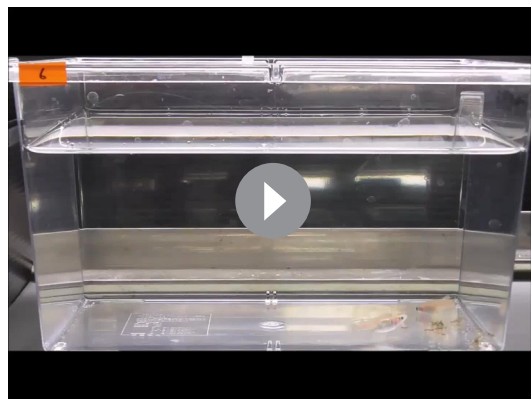

**Video 3.** A representative video showing the mating behavior of *npbwr2* knockout females.
DOI: https://doi.org/10.7554/eLife.39495.024

as gonadal estrogen might contribute to the female-specific expression of *npba* in medaka brain. Here, however, we found that the sexually dimorphic pattern of *npba* expression is independent of sex chromosome complement, and relies exclusively on gonadal sex steroids. The lack of sex chromosome effects certainly underlies the reversibility of the sexually dimorphic *npba* expression.

We also showed that the female-specific expression of *npba* results, at least in part, from direct transcriptional activation of *npba* by estrogen via an ERE present in its promoter. Consistent with this, female-specific *npba*-expressing neurons coexpress estrogen receptors (ERs) (*Hiraki et al., 2014*). The identification of *npba* as a direct target of estrogen action is interesting because very few genes involved in female mating behavior in any species have been found to be direct targets of estrogen, although such genes can be highly estrogen-responsive (*Yang and Shah, 2014*). The exceptions are the genes encoding the progesterone receptor, oxytocin, and oxytocin receptor; however, those genes are expressed in behaviorally relevant brain regions of both males and females, and affect the mating behaviors of both sexes, in contrast to *npba* in medaka (*Wagner, 2006*; *Yang et al., 2013*; *Veening et al., 2015*; *Dumais and Veenema, 2016*; *Pfaff et al., 2018*). Collectively, our findings indicate that the female-specific expression of *npba* has a distinct regulatory mechanism.

Another distinctive aspect of *npba* expression is the bidirectional reversibility of sexual dimorphism (both from female-typical to male-typical pattern and from male-typical to female-typical pattern). We found that *npba* expression in Vs/Vp and PMm/PMg disappeared in females and appeared in males in response to changes in the adult sex steroid milieu, in contrast to genes that exhibit sexually dimorphic expression and mediate sex-typical mating behaviors in mammals. In mice, for example, the cholecystokinin receptor gene (*Cckar*) implicated in female sexual receptivity is expressed in the ventromedial nucleus of the hypothalamus at much higher levels in females owing to estrogen stimulation. Although *Cckar* expression in this nucleus is reduced by ovariectomy and reinstated by treatment with estrogen in adult females, treating adult males with estrogen does not induce expression of *Cckar* in this nucleus (*Xu et al., 2012*). Such irreversibility of sexually dimorphic gene expression in the adult brain is attributed to the specific action of gonadal sex steroids during the perinatal period, in addition to the sex chromosome effects mentioned above. Studies in rodents indicate that perinatal sex steroids have long-lasting or even permanent effects on brain gene expression and behavior via epigenetic mechanisms involving DNA methylation and histone modifications, which establish and maintain differences between the sexes across the lifespan (*McCarthy and Nugent, 2015*; *Forger et al., 2016*; *McCarthy et al., 2017*). The reversible sex difference in adult *npba* expression in medaka suggests either that perinatal sex steroids induce few, if any, epigenetic modifications in *npba* or that such modifications can easily be reversed for some reason. This unique mode of sexual differentiation of *npba* expression probably contributes to the adult reversal of sex-typical mating behaviors observed in teleosts. Given that mammals and birds also exhibit some degree of sexual lability in mating behavior, behavior-relevant genes for which expression in the adult brain is sexually dimorphic but bidirectionally reversible in response to sex steroids may remain to be identified.

Importantly, our results provide insight into the mechanisms underlying the reversal of sexually dimorphic *npba* expression. The profound decrease in *npba* expression observed in ovary-intact females treated with non-aromatizable androgen indicates that androgen/androgen receptor (AR) signaling plays an inhibitory role in *npba* expression, alongside the stimulatory role of estrogen/ER signaling. However, expression of *npba* was not induced in males deprived of androgen by castration, suggesting that the androgen/AR signaling does not directly inhibit *npba* expression. Considering a concomitant decrease in brain levels of E2 in androgen-treated males, the androgen/AR

signaling pathway presumably exerts its influence by attenuating the stimulatory action of estrogen through a reduction of brain E2 levels. Given that ER expression in Vs/Vp and PMp/PMg is strongly suppressed by androgen (*Hiraki et al., 2012*), the inhibitory action of androgen on *npba* expression may involve repression of estrogen/ER signaling not only at ligand, but also at receptor level.

The estrogenic induction of *npba* expression in Vs/Vp and PMm/PMg of males showed a slower kinetic response than would be expected of a model of direct transcriptional activation. However, we previously found that ERs are not expressed at detectable levels in Vs/Vp and PMm/PMg of males and that treating males with estrogen induces ER expression therein (*Hiraki et al., 2012*). Considering these findings, it seems plausible that this estrogen-induced *npba* expression is not fully realized until sufficient ER expression is achieved.

The brains of teleosts exhibit extensive adult neurogenesis (*Chapouton et al., 2007*; *Ganz and Brand, 2016*) and estrogen has significant effects on adult neurogenesis (*Mahmoud et al., 2016*; *Heberden, 2017*; *Ponti et al., 2018*). These facts suggest the possibility that adult neurogenesis may be involved in the reversal of sexually dimorphic *npba* expression. However, our recent work showed that the Npba-expressing neurons emerged in adult males treated with estrogen, not through neurogenesis, but through the activation of pre-existing, quiescent male counterpart neurons (*Kikuchi et al., 2019*). This evidence suggests the relevance of an activation–inactivation process on existing neurons, rather than neurogenesis, in the reversible sexual dimorphism in *npba* expression.

We found that the female-specific *npba*-expressing neurons project their axons to various regions of the central nervous system, particularly the brainstem and spinal cord, but not to the pituitary. The latter observation is surprising, given that most of the isotocin neurons that are intermingled with the female-specific *npba*-expressing neurons in PMm/PMg project to the pituitary in teleosts including medaka (*Hiraki et al., 2014*; *Yamashita et al., 2017*). Consistent with the widespread projection of *npba*-expressing neurons, *npbwr2* was found to be widely expressed in the central nervous system, corroborating the idea that the female-specific Npba peptide has multiple action sites. In the spinal cord, *npbwr2* was expressed in the medial part of the dorsal horn and the lateral part of the ventral horn, which respectively contain sympathetic preganglionic neurons and motoneurons in teleosts as well as in mammals (*Westerfield et al., 1986*; *Schneider and Sulner, 2006*; *Funakoshi and Nakano, 2007*). Thus, the female-specific Npba probably acts on either or both of these neurons in the spinal cord, as well as in various neuronal subsets in the brain. Although NPBWR1/2 primarily couples to the inhibitory G protein (Gi) (*Fujii et al., 2002*; *Tanaka et al., 2003*), activation of NPBWR1 reportedly leads to increased excitability of some neuronal populations in rats (*Price et al., 2009*). Thus, whether Npbwr2 activation has inhibitory or stimulatory effects on each type of neuronal population in medaka remains to be defined.

Our study identified a second functional NPB gene, designated *npbb*, in medaka. Phylogenetic and syntenic analyses indicated that this gene probably originated from the teleost-specific whole-genome duplication (*Amores et al., 1998*) and is thus confined to teleosts. Similar to *npba*, *npbb* was found to be preferentially expressed in females in Vs/Vp and PMm/PMg/PPp; in addition, *npba* and *npbb* were expressed in the same neurons in these brain nuclei, suggesting that the two genes share common regulatory mechanisms. Indeed, the sexually dimorphic expression of *npbb* in Vs/Vp and PMm/PMg/PPp could also be reversed by altering the sex steroid milieu, although the changes were less marked as compared with *npba*. In addition, a putative ERE, though not yet proven to be functional, was found in the *npbb* promoter at a similar position as in the *npba* promoter, suggesting that *npbb* is also directly activated by estrogen and that a cis-regulatory region that confers estrogen-responsive NPB gene expression is evolutionary conserved. This idea is further supported by the observation that NPB genes in gar and several other teleosts also contain a putative ERE at a similar position. The results also suggested that the two genes have a shared function in female mating behavior; indeed, part of the altered behavioral phenotypes was evident in *npba*$^{-/-}$/*npbb*$^{-/-}$ females, but not in single knockouts of either of the two genes, presumably due to functional redundancy. Importantly, coexpression of *npba* and *npbb* was observed exclusively in Vs/Vp and PMm/PMg/PPp and not in other brain nuclei, which strongly suggests that their expression in these nuclei is responsible for the phenotypes observed.

*npba*$^{-/-}$, *npba*$^{-/-}$/*npbb*$^{-/-}$, and *npbwr2*$^{-/-}$ females showed increased latency from courtship display to wrapping, and furthermore, *npba*$^{-/-}$/*npbb*$^{-/-}$ and *npbwr2*$^{-/-}$ females showed increased incidence of spawning without any preceding courtship display. These results illustrate the relevance of NPB

signaling to female sexual receptivity. However, the two phenotypes are seemingly inconsistent in that the increased latency from courtship to mating may be associated with decreased receptivity, whereas the increased incidence of spawning without receiving courtship may be associated with increased receptivity. The most likely explanation is that the deficiency of NPB signaling reduces the significance of courtship display in female receptivity. In medaka, as in other animals, courtship display serves to stimulate female receptivity (*Grant et al., 1995*; *Grant and Green, 1996*). Conceivably, courtship display is somehow less important and less effective for females deficient in NPB signaling; in other words, they require more time to accept males after receiving courtship stimulation, but simultaneously tend to accept males without being courted. Thus, female-specific NPB signaling may play an important role in female mate choice by facilitating the acceptance of males performing courtship display and the refusal of males exhibiting no courtship.

This raises the question of what specific process in female mate choice is primarily influenced by female-specific NPB signaling. The answer remains to be found, but $Npbwr1^{-/-}$ mice have been shown to exhibit behavioral abnormalities in social interaction: when confronted by a conspecific intruder, they approach rapidly and spend more time close to the intruder (*Nagata-Kuroiwa et al., 2011*). In addition, a single nucleotide polymorphism in human *NPBWR1* has been shown to influence the evaluation of facial expressions (*Watanabe et al., 2012*). These observations, together with the extensive effective area of female-specific NPB signaling in the brain, suggest that NPB signaling may be involved in the process of perceiving and/or evaluating male performance during courtship. Alternatively, given that the spinal cord is the major target site of female-specific NPB signaling, it is possible that NPB signaling, by influencing motor and/or sympathetic function, may be involved in executing the decision of whether or not to mate.

In summary, the present study has identified NPB as a critical element of the female sexual receptivity circuitry in medaka, exhibiting reversible sexual dimorphism under the direct control of estrogen. The female-specific NPB neurons are located in Vs/Vp and PMm/PMg, regions that are considered homologous to the bed nucleus of the stria terminalis/subpallial amygdala and the supraoptic nucleus/paraventricular nucleus, respectively, in the mammalian brain (*Northcutt, 1995*; *Moore and Lowry, 1998*; *O'Connell and Hofmann, 2011*; *Goodson and Kingsbury, 2013*). Similar to medaka, NPB/NPW neurons have been detected in these nuclei in mammals (*Dun et al., 2003*; *Jackson et al., 2006*). Thus, the role of NPB in female sexual receptivity may be conserved across a wide array of vertebrate taxa. Further studies will be required to test this possibility.

## Materials and methods

**Key resources table**

| Reagent type (species) or resource | Designation | Source or reference | Identifiers | Additional information |
|---|---|---|---|---|
| Gene (*Oryzias latipes*) | *npba* | DOI: 10.1210/en.2013-1806 | Genbank:NM_001308979 | |
| Gene (*O. latipes*) | *npbb* | NBRP Medaka | clone ID:olova57o22; clone ID:olova6h09; clone ID:olova8d12; clone ID:olova13m04; clone ID:olova28d23; clone ID:olova58i17 | |
| Gene (*O. latipes*) | *npbwr2* | | Genbank:LC375958 | |
| Gene (*O. latipes*) | *actb* | | Genbank:NM_001104808 | |
| Strain, strain background (*O. latipes*) | d-rR | NBRP Medaka | strain ID:MT837 | maintained in a closed colony over 10 years in Okubo lab |
| Genetic reagent (*O. latipes*) | ΔERE mutant | this paper | | generated and maintained in Okubo lab |
| Genetic reagent (*O. latipes*) | *npba*-GFP transgenic | this paper | | generated and maintained in Okubo lab |

*Continued on next page*

*Continued*

| Reagent type (species) or resource | Designation | Source or reference | Identifiers | Additional information |
|---|---|---|---|---|
| Genetic reagent (*O. latipes*) | *npba*<sup>-/-</sup> | this paper | | generated and maintained in Okubo lab |
| Genetic reagent (*O. latipes*) | *npbb*<sup>-/-</sup> | this paper | | generated and maintained in Okubo lab |
| Genetic reagent (*O. latipes*) | *npbwr2*<sup>-/-</sup> | this paper | | generated and maintained in Okubo lab |
| Cell line (*Homo sapiens*) | HEK293T | Riken BRC Cell Bank | Cell number:RCB2202; RRID:CVCL_0063 | |
| Cell line (*Escherichia coli*) | DY380 | DOI: 10.1038/35093556; DOI: 10.1006 /geno.2000.6451 | | |
| Transfected construct | pcDNA3.1/V5-His-TOPO | Thermo Fisher Scientific | Thermo Fisher Scientific:K480001 | |
| Transfected construct | pGL4.29 | Promega | Promega:E8471 | |
| Transfected construct | pGL4.74 | Promega | Promega:E6921 | |
| Antibody | anti-Npba antibody | this paper | RRID:AB_2810229 | rabbit polyclonal; against the entire Npba polypeptide (1:500 or 1:1000) |
| Antibody | Dylight 549-conjugated goat anti-rabbit IgG | Vector Laboratories | Vector Laboratories:DI-1549; RRID:AB_2336407 | (1:500 or 1:1000) |
| Antibody | Alexa Flour 488-conjugated goat anti-rabbit IgG | Thermo Fisher Scientific | Thermo Fisher Scientific:A-11070; RRID:AB_2534114 | (1:500) |
| Antibody | horseradish peroxidase-conjugated anti-fluorescein antibody | PerkinElmer | PerkinElmer: NEF710001EA; RRID:AB_2737388 | (1:500 or 1:1000) |
| Antibody | alkaline phosphatase-conjugated anti-DIG antibody | Roche Diagnostics | Roche Diagnostics: 1109327491 0; RRID:AB_514497 | (1:500–1:10000) |
| Recombinant DNA reagent | medaka bacterial artificial chromosome (BAC) clone | NBRP Medaka | clone ID:180_I09 | |
| Recombinant DNA reagent | phrGFP II-1 mammalian expression vector | Agilent Technologies | Agilent Technologies:240143 | |
| Recombinant DNA reagent | pGEM-Teasy vector | Promega | Promega:A1360 | |
| Peptide, recombinant protein | Npba polypeptide | this paper | | WYKQVAGPSYYSVGR ASGLLSGIRRSPHV-NH2 |
| Peptide, recombinant protein | Npbb polypeptide | this paper | | WYKQSTGPIFYPVGRA SGLLSGIRRSPYV-NH2 |
| Commercial assay or kit | Dual-Luciferase Reporter Assay System | Promega | Promega:E1910 | |
| Commercial assay or kit | RNeasy Lipid Tissue Mini Kit | Qiagen | Qiagen:74804 | |
| Commercial assay or kit | RNeasy Plus Universal Mini Kit | Qiagen | Qiagen:73404 | |

*Continued on next page*

*Continued*

| Reagent type (species) or resource | Designation | Source or reference | Identifiers | Additional information |
|---|---|---|---|---|
| Commercial assay or kit | Omniscript RT Kit | Qiagen | Qiagen:205111 | |
| Commercial assay or kit | SuperScript VILO cDNA Synthesis Kit | Thermo Fisher Scientific | Thermo Fisher Scientific:11754050 | |
| Commercial assay or kit | LightCycler 480 SYBR Green I Master | Roche Diagnostics | Roche Diagnostics: 04887352001 | |
| Commercial assay or kit | Golden Gate TALEN and TAL Effector Kit 2.0 | Addgene | Addgene:1000000024 | |
| Commercial assay or kit | mMessage mMachine SP6 Kit | Thermo Fisher Scientific | Thermo Fisher Scientific:AM1340 | |
| Commercial assay or kit | Marathon cDNA Amplification Kit | Takara Bio | Takara Bio:634913 | |
| Commercial assay or kit | Power SYBR Green PCR Master Mix | Thermo Fisher Scientific | Thermo Fisher Scientific:4367659 | |
| Commercial assay or kit | TSA Plus Fluorescein System | PerkinElmer | PerkinElmer: NEL741001KT | |
| Chemical compound, drug | aromatase inhibitor (AI); Fadrozole | Sigma-Aldrich | Sigma-Aldrich: F3806-10MG | |
| Chemical compound, drug | estradiol-17β; E2 | Fujifilm Wako Pure Chemical Corporation | Fujifilm Wako Pure Chemical Corporation: 058–04043 | |
| Chemical compound, drug | 11-ketotestosterone; KT | Cosmo Bio | Cosmo Bio:117 ST | |
| Software, algorithm | GraphPad Prism | GraphPad Software | RRID:SCR_002798 | |
| Software, algorithm | Adobe Photoshop | Adobe Systems | RRID:SCR_014199 | |
| Software, algorithm | ImageJ | http://rsbweb.nih.gov/ij/ | RRID:SCR_003070 | |
| Software, algorithm | InterProScan | http://www.ebi.ac.uk/ interpro/search/ sequence-search | RRID:SCR_005829 | |
| Software, algorithm | SignalP | http://www.cbs.dtu.dk/ services/SignalP/ | RRID:SCR_015644 | |
| Software, algorithm | ClustalW | http://clustalw.ddbj. nig.ac.jp/index.php | RRID:SCR_017277 | |
| Software, algorithm | Jaspar | http://jaspar.genereg.net/ | RRID:SCR_003030 | |
| Other | DAPI stain | Thermo Fisher Scientific | Thermo Fisher Scientific:D1306; RRID:AB_2629482 | (1:1000) |

## Animals

Medaka of the d-rR strain were bred and maintained at 28°C under a 14 hr light/10 hr dark photoperiod. They were fed 3–4 times per day with live brine shrimp and commercial pellet food (Otohime; Marubeni Nissin Feed, Tokyo, Japan). Sexually mature, spawning adults (aged 3–5 months) were used for all analyses. They were randomly assigned to experimental groups. All sampling was conducted at 1–4 hr after onset of the light period.

## Production of sex-reversed medaka

Sex-reversed XX males and XY females were produced as described previously (*Okubo et al., 2011*). In brief, fertilized eggs were treated with methyltestosterone at high temperature (32°C) for the production of XX males, or with E2 at 28°C for the production of XY females.

## Real-time PCR

The brain was sampled whole (excluding the cerebellum and medulla oblongata in ΔERE mutant analysis) or dissected into three parts (the olfactory bulb, telencephalon, diencephalon, and midbrain tegmentum; the optic tectum; and the cerebellum and medulla oblongata), and total RNA was isolated by using the RNeasy Lipid Tissue Mini Kit or RNeasy Plus Universal Mini Kit (Qiagen, Hilgen, Germany). Complementary DNA was synthesized by using the Omniscript RT Kit (Qiagen) or Super-Script VILO cDNA Synthesis Kit (Thermo Fisher Scientific, Waltham, MA). Real-time PCR was run either on the ABI Prism 7000 Sequence Detection System using the Power SYBR Green PCR Master Mix (Thermo Fisher Scientific) or on the LightCycler 480 System II using the LightCycler 480 SYBR Green I Master (Roche Diagnostics, Basel, Switzerland). For every reaction, melting curve analysis was conducted to ensure that a single amplicon was produced in each sample. The β-actin gene (*actb*; GenBank accession number NM_001104808) was used to normalize levels of *npba*/*npbb* transcripts in each sample. The primers used for real-time PCR are listed in *Supplementary file 2*. The numbers of fish examined in each experimental group are indicated in the figure legends.

## Single in situ hybridization

DNA fragments of 645 bp, 647 bp, and 1520 bp corresponding to nucleotides 1–645 of the medaka *npba* cDNA (GenBank accession number NM_001308979), 1–647 of the medaka *npbb* EST clone of National BioResource Project (NBRP) Medaka (http://www.shigen.nig.ac.jp/medaka/) (clone ID: olo-va57o22), and 1699–3218 of the medaka *npbwr2* cDNA (deposited in DDBJ/EMBL/GenBank with accession number LC375958), respectively, were PCR-amplified and subcloned into the pGEM-Teasy vector (Promega, Madison, WI). The resultant constructs were used to generate digoxigenin (DIG)-labeled cRNA probes for *npba*, *npbb*, and *npbwr2* by using the DIG RNA Labeling Mix and T7 RNA polymerase (Roche Diagnostics). The procedure for in situ hybridization has been described previously (*Hiraki et al., 2012*). In brief, the brain, pituitary, and/or spinal cord were fixed in 4% paraformaldehyde (PFA), embedded in paraffin, and cut into 10 μm sections in the coronal plane. Hybridization signals were visualized by using alkaline phosphatase-conjugated anti-DIG antibody (Roche Diagnostics) and 5-bromo-4-chloro-3-indolyl phosphate/nitro blue tetrazolium (BCIP/NBT) substrate (Roche Diagnostics). Color development was allowed to proceed overnight or was stopped after 2–3 hr (for quantification of *npba* expression in Vs/Vp) or 15–30 min (for quantification of *npba* expression in PMm/PMg) to avoid saturation. All sections in each comparison were processed simultaneously under the same conditions. The numbers of fish examined in each experimental group are indicated in the figure legends.

To obtain quantitative data, the number of *npba*/*npbb*-expressing neurons in each brain nucleus or each brain/spinal cord region was counted manually. In some analyses, the resulting sections were photographed and converted to black and white binary images by thresholding using Adobe Photoshop (Adobe Systems, San Jose, CA), and the total area of *npba*/*npbb* expression was calculated by using ImageJ (http://rsbweb.nih.gov/ij/). The threshold was set manually by visual inspection of the images to successfully separate specific signals from background noise. Brain nuclei were identified by using medaka brain atlases (*Anken and Bourrat, 1998*; *Ishikawa et al., 1999*; http://www.shigen.nig.ac.jp/medaka/medaka_atlas/), supplemented with information from our Nissl-stained sections (*Kawabata et al., 2012*).

## Generation of the ere mutant medaka

Mutant medaka in which an ERE sequence in the *npba* promoter was deleted (ΔERE) were generated by TALEN-mediated genome editing. TALENs were designed to target the ERE that has been shown to be functional in vitro (*Hiraki et al., 2014*). The position and sequence of the targeted ERE and TALEN binding sites are shown in *Figure 1—figure supplement 1*. The TALE repeat arrays were assembled by the Golden Gate method (*Cermak et al., 2011*) as described elsewhere (*Ansai et al., 2013*), using the Golden Gate TALEN and TAL Effector Kit 2.0 (Addgene 1000000024). TALEN mRNAs were synthesized by in vitro transcription using the mMessage mMachine SP6 Kit (Thermo Fisher Scientific) and microinjected into the cytoplasm of embryos at the one-cell stage. Potential founders were screened by outcrossing to wild-type fish and testing progeny for mutations at the target site using a mismatch-sensitive T7 endonuclease I assay (*Kim et al., 2009*) followed by direct sequencing. A founder was identified that yielded a high proportion of progeny carrying a 7

bp deletion; the progeny were subsequently intercrossed to generate fish homozygous for the deletion. The genotype of each fish was determined by direct sequencing using the primers listed in *Supplementary file 2*. In all analyses of ΔERE mutants, their wild-type siblings were used as controls.

## Gonadectomy and drug treatment

For both male and female fish, the gonad was removed under tricaine methane sulfonate anesthesia (0.02%) through a small incision made in the ventrolateral abdominal wall. Immediately after removal of the gonad, the incision was sutured with nylon thread. Sham-operated fish received the same surgical treatment as gonadectomized fish, except for removal of the gonad. After a recovery period of 3 days, gonadectomized males and females were immersed in water containing 100 ng/ml of KT or E2, or vehicle (ethanol) alone for 6 days and then sampled. Sham-operated fish were treated with vehicle alone and used as controls.

In another experiment, females with intact ovaries were treated with 100 ng/ml of KT or AI (Fadrozole; Sigma-Aldrich, St. Louis, MO) by immersion in water for 9 days. Similarly, males with intact testes were treated with 100 ng/ml of E2 by immersion in water for 9 days. These fish were sampled on days 0 (untreated controls), 2, 5, and 9. The sex steroid concentration used was based on previously reported serum steroid levels in medaka (*Foran et al., 2002*; *Foran et al., 2004*; *Tilton et al., 2003*).

## Measurement of brain levels of E2

Dissected whole brains were frozen and stored at −80˚C until analysis. Levels of E2 in the brains were analyzed by liquid chromatography coupled to tandem mass spectrometry (LC-MS/MS) at Aska Pharmamedical (Kanagawa, Japan). In the KT treatment experiment, E2 levels were measured over the same time course as that used to assess *npba/npbb* expression (on days 0 (untreated controls), 2, 5, and 9 of treatment).

## Double labeling with immunohistochemistry and in situ hybridization

A polyclonal anti-Npba antibody was raised in rabbit against a synthetic peptide corresponding to the entire mature polypeptide sequence of medaka Npba (Medical and Biological Laboratories, Aichi, Japan). The specificity of the anti-Npba antibody in immunohistochemical studies was confirmed by a double-labeling experiment with immunohistochemistry and in situ hybridization. In brief, whole brains of females (n = 6) were fixed in 4% PFA and embedded in 5% agarose (Type IX-A; Sigma-Aldrich) supplemented with 20% sucrose. Frozen coronal sections of 20 μm thickness were cut and hybridized with the above-mentioned *npba* probe, which was labeled with fluorescein by using Fluorescein RNA Labeling Mix and T7 RNA polymerase (Roche Diagnostics). After blocking with PBS containing 2% normal goat serum for 30 min, the sections were incubated overnight at 4˚C with the anti-Npba antibody diluted 1:500 in phosphate-buffered saline (PBS) containing 2% normal goat serum, 0.1% bovine serum albumin, and 0.02% keyhole limpet hemocyanin. The sections were incubated overnight at 4˚C with horseradish peroxidase-conjugated anti-fluorescein antibody (diluted 1:1000 and 1:500 for detection in PMm/PMg and other nuclei, respectively; PerkinElmer, Waltham, MA) and Dylight 549-conjugated goat anti-rabbit IgG (diluted 1:500; Vector Laboratories, Burlingame, CA) in Tris-buffered saline (TBS) containing 1.5% blocking reagent (Roche Diagnostics) and 5 μg/ml 4′,6-diamidino-2-phenylindole (DAPI). The anti-fluorescein antibody was visualized by using the TSA Plus Fluorescein System (PerkinElmer) in accordance with the manufacturer's instructions. Fluorescent images were acquired by a confocal laser scanning microscope (C1; Nikon, Tokyo, Japan). The following excitation and emission wavelengths were used for detection: DAPI for nuclear staining, 405 nm and 450/35 nm; fluorescein, 488 nm and 515/30 nm; Dylight 549, 543 nm and 605/75 nm.

## Immunohistochemistry

Immunohistochemistry was performed with the anti-Npba antibody described above. Whole brains, with pituitaries attached, and spinal cords of both sexes (n = 3 each) were fixed in 4% PFA and embedded in 5% agarose (Type IX-A; Sigma-Aldrich) supplemented with 20% sucrose. Frozen 40 μm thick sections were cut in the sagittal plane. After blocking as described above, the sections

were incubated overnight at 4°C with the anti-Npba antibody diluted at 1:1000 in PBS containing 2% normal goat serum, 0.1% bovine serum albumin, and 0.02% keyhole limpet hemocyanin. The sections were incubated overnight at 4°C with Dylight 549-conjugated goat anti-rabbit IgG (diluted 1:1000; Vector laboratories) or Alexa Fluor 488-conjugated goat anti-rabbit IgG (diluted 1:500; Thermo Fisher Scientific) in PBS. Fluorescent images were obtained as described above. The excitation and emission wavelengths for Alexa Fluor 488 were 488 nm and 515/30 nm, respectively.

## Generation of the *npba*-GFP transgenic medaka

A medaka bacterial artificial chromosome (BAC) clone (clone ID: 180_I09) containing the *npba* locus was obtained from NBRP Medaka and modified by homologous recombination in *Escherichia coli* strain DY380, essentially as described previously (*Copeland et al., 2001*; *Lee et al., 2001*). A 22 bp sequence containing the translation initiation site of *npba* in this BAC clone was replaced by a 2136 bp DNA cassette containing the humanized *Renilla reniformis* GFP II-coding sequence (Agilent Technologies, Santa Clara, CA), bovine growth hormone polyadenylation signal, and kanamycin resistance gene (*Figure 3—figure supplement 2*). The resulting BAC transgene was microinjected into the cytoplasm of embryos at the one-cell stage. Transgenic founders were screened by outcrossing to wild-type fish and examining progeny embryos for GFP fluorescence. Two founders were identified that produced progeny expressing GFP in a pattern reflective of endogenous *npba* expression during embryonic development (*Hiraki et al., 2014*). These progeny were raised to adulthood and intercrossed to establish homozygous transgenic lines.

## Double in situ hybridization

To confirm that GFP expression in the *npba*-GFP transgenic fish recapitulated endogenous *npba* expression, brains from transgenic fish (n = 10; 3 males and seven females) were processed for double in situ hybridization using a fluorescein-labeled GFP probe and a DIG-labeled *npba* probe. Double in situ hybridization was also performed on wild-type female brains (n = 5) with a fluorescein-labeled *npba* probe and a DIG-labeled *npbb* probe, in order to examine whether *npba* and *npbb* are coexpressed in the same neurons. Double in situ hybridization was done according to *Takeuchi and Okubo (2013)*. In brief, whole brains were fixed in 4% PFA and embedded in 5% agarose (Type IX-A; Sigma-Aldrich) supplemented with 20% sucrose. Frozen 20 µm thick coronal sections were cut and hybridized with the fluorescein- and DIG-labeled probes. The GFP probe was generated from a 733 bp fragment corresponding to nucleotides 733–1465 of the phrGFP II-1 mammalian expression vector (Agilent Technologies). The *npba* and *npbb* probes used were the same as those used for single in situ hybridization. The fluorescein-labeled probe was visualized by using the TSA Plus Fluorescein System (PerkinElmer); the DIG-labeled probe was visualized by using alkaline phosphatase-conjugated anti-DIG antibody (Roche Diagnostics) and Fast Red (Roche Diagnostics). Cell nuclei were counterstained with DAPI. Fluorescent images were obtained as described above.

## GFP imaging

Wide-field fluorescence imaging was performed on whole, unsectioned brains and spinal cords of *npba*-GFP transgenic fish (n = 2 for each sex). Whole brains, with pituitaries attached, and spinal cords were fixed in 4% PFA and optically cleared by immersion in Scaleview-A2 (Olympus, Tokyo, Japan) at room temperature overnight. Confocal images were acquired by using a fluorescence macroscope (MVX10; Olympus) equipped with a DSU spinning disk confocal system and EGFP band pass filter set (Olympus).

## Molecular cloning of medaka *npbwr2*

An EST library was generated from the medaka brain and approximately 32000 ESTs were randomly selected and sequenced in the 5′ to 3′ direction as described previously (*Okubo et al., 2011*). After assembly and annotation, an EST (clone ID: 11_L07), which had best BLAST hits to *NPBWR1/NPBWR2* in other vertebrates, was identified and fully sequenced. Upon sequencing, this EST was found to be truncated at the 5′ end and to lack the translation initiation site. The remaining 5′ sequence was obtained by 5′-RACE on medaka brain poly(A)$^+$ RNA using the Marathon cDNA Amplification Kit (Takara Bio, Shiga, Japan), essentially as described previously (*Kawabata et al., 2012*).

The deduced amino acid sequence of the resultant full-length medaka cDNA was aligned with NPBWR1/NPBWR2 in other vertebrates by using ClustalW. The resulting alignment was used to construct a bootstrapped (1000 replicates) neighbor-joining tree (http://clustalw.ddbj.nig.ac.jp/index.php). Opioid receptors μ1 and δ1 (OPRD1 and OPRM1) in humans were used as outgroups. The species names and GenBank accession numbers of the sequences used in the analysis are listed in *Supplementary file 3*.

## Molecular cloning of medaka *npbb*

A survey of the medaka genome assembly in Ensembl (http://www.ensembl.org/index.html) identified a previously uncharacterized predicted gene (gene ID: ENSORLG00000012098) that bears some sequence similarity to *npba*. A BLAST search of the medaka EST database at NBRP using this predicted gene, designated *npbb*, as the query identified the corresponding ESTs (clone ID: olova57o22, olova6h09, olova8d12, olova13m04, olova28d23, olova58i17), which were then assembled to obtain the full-length cDNA sequence for *npbb*.

The deduced amino acid sequence of medaka *npbb* was analyzed for the presence of specific domains or motifs by using InterProScan (http://www.ebi.ac.uk/interpro/search/sequence-search) and SignalP (http://www.cbs.dtu.dk/services/SignalP/). This sequence was then aligned with NPB/NPW in other vertebrates by using ClustalW, and an unrooted neighbor-joining tree with 1000 bootstrap replicates was constructed (http://clustalw.ddbj.nig.ac.jp/index.php). The species names and GenBank accession numbers of the sequences used are listed in *Supplementary file 3*.

Syntenic relationships of genes in the vicinity of gar, arowana, catfish, and medaka NPB genes were established using the Ensembl genome browser. The proximal promoter sequence of medaka *npbb* (up to 6 kb upstream of the translation initiation site) was retrieved from the Ensembl genome browser and analyzed for the presence of putative EREs by Jaspar (http://jaspar.genereg.net/). Only putative EREs with a relative profile score higher than 80% on both strands were considered as positive hits. The proximal promoter sequences of gar, arowana, catfish, and tilapia NPB genes were also obtained and analyzed for putative EREs by the same procedure.

## Receptor activation assay

Medaka Npba and Npbb polypeptides with amidated C termini were synthesized by Scrum (Tokyo, Japan). The cDNA fragment encoding the full-length Npbwr2 was PCR-amplified and subcloned into the expression vector pcDNA3.1/V5-His-TOPO (Thermo Fisher Scientific). The resulting Npbwr2 expression construct was transiently transfected into HEK293T cells together with the cAMP-responsive luciferase reporter vector pGL4.29 (Promega) and the internal control vector pGL4.74 (Promega) at a ratio of 11:18:1 using FuGENE HD transfection reagent (Promega). Forty-two hours after transfection, cells were stimulated with peptide at doses of 0, $10^{-11}$, $10^{-10}$, $10^{-9}$, $10^{-8}$, $10^{-7}$, $10^{-6}$, and $10^{-5}$ M in the presence of 2 μM forskolin for 6 hr. The extracted cells were assayed for luciferase activity by using the Dual-Luciferase Reporter Assay System (Promega). Assays were performed in triplicate and repeated twice independently. HEK293T cells used in this study were authenticated by short tandem repeat profiling (National Institute of Biomedical Innovation, Osaka, Japan) and confirmed to be mycoplasma free (Biotherapy Institute of Japan, Tokyo, Japan).

## Generation of knockout medaka

Knockout medaka deficient for *npba*, *npbb*, or *npbwr2* were generated by using TALEN technology, targeting the sequences corresponding to the N-terminus of the mature Npba/Npbb polypeptide and the middle portion of Npbwr2 protein (*Figure 6—figure supplement 1*). TALEN construction, mRNA synthesis, microinjection, and founder screening were performed as described above. A founder was identified for each knockout strain that yielded a high proportion of progeny carrying a deletion that caused a frameshift and premature termination of translation (10 bp, 11 bp, and 7 bp deletions for *npba*-/-, *npbb*-/-, and *npbwr2*-/- strains, respectively) (*Figure 6—figure supplement 1*). These progeny were intercrossed to generate fish homozygous for the deletions. The genotype of each fish was determined by direct sequencing using the primers listed in *Supplementary file 2*. In addition, *npba/npbb* double knockout (*npba*-/-/*npbb*-/-) medaka were obtained by intercrossing the *npba*-/- and *npbb*-/- medaka.

## Mating behavior analysis

All behavioral procedures were conducted in 2-liter rectangular tanks contained within a large recirculating water system with a constant influx of dechlorinated tap water. Fish were thinned to 2–4 females and 2–4 males per tank and allowed to acclimatize for 3–5 days to ensure and standardize their reproductive conditions. On the day before behavioral testing, each focal female was placed with a stimulus male in a tank, separated by a transparent, perforated partition. The partition was removed 1.0–2.5 hr after onset of the light period of the following day, and fish were allowed to interact for 30 min. All interactions were recorded with a digital video camera (iVIS HF S11/S21, Canon, Tokyo, Japan, or Everio GZ-G5, Jvckenwood, Kanagawa, Japan). Water inflow to the tanks was turned off during recording.

The percentage of females that spawned within the test period (30 min) was calculated for each genotype. The following behavioral parameters were also calculated from the video recordings: the total number of courtship displays; the latency from the first courtship display to the first wrapping and to the wrapping that resulted in spawning; the number of wrapping attempts refused by the female; the duration of quivering; and the percentage of females that spawned without any preceding courtship display from the male. Each action during mating behavior was identified following *Ono and Uematsu (1957)* and *Walter and Hamilton (1970)* (also see a brief description of each action in the Results section). Females that did not spawn were excluded from the analysis of these parameters. Females that spawned without any courtship display were excluded from the analysis of latency from the first courtship display.

In the behavioral analysis of *npba* single knockouts, their wild-type and heterozygous siblings served as controls. In the analysis of *npbb* single knockouts, their heterozygous siblings and non-sibling wild-type fish served as controls, because *npbb* is located on the sex chromosomes and it is impossible to obtain wild-type siblings of *npbb* knockouts by the mating of any pair of genotypes. For the same reason, non-sibling wild-type fish were used as controls in the analysis of *npba/npbb* double knockouts and *npbwr2* knockouts. These non-sibling wild-type fish were derived from the same genetic background and reared under the same conditions as the knockouts.

## Statistical analysis

Values are presented as mean ± standard error of the mean (SEM) for continuous variables and as percentages for categorical variables. Individual data points are also shown to give a better indication of the underlying distribution. To facilitate comparisons in real-time PCR analysis, the expression level of each target gene (normalized to that of *actb*) in male whole brain was arbitrarily set to 1, and the relative difference was calculated.

Statistical analyses were performed by using GraphPad Prism (GraphPad Software, San Diego, CA). Continuous variables were compared between two groups by the unpaired two-tailed Student's *t*-test. If the F-test indicated that the variance differed significantly between groups, Welch's correction to the Student's *t*-test was employed. For more than two groups, continuous variables were compared by one-way analysis of variance (ANOVA), followed by either Bonferroni's (for comparisons among experimental groups) or Dunnett's (for comparisons of experimental versus control groups) *post hoc* test. If Bartlett's and Brown-Forsythe tests indicated that the variance differed significantly among groups, data were log-transformed to normalize distributions prior to analysis. If the variance remained heterogeneous after transformation, data were analyzed by the non-parametric Kruskal-Wallis test followed by Dunn's *post hoc* test. For the analysis of *npba* expression in ovariectomized and E2-treated ΔERE mutant females, a two-way ANOVA was conducted to test for main effects and interactions between genotype and treatment. Behavioral time-series data sets were further analyzed using Kaplan-Meier plots with the inclusion of fish that did not spawn within the test period, following *Jahn-Eimermacher et al. (2011)*. Differences between Kaplan-Meier curves were tested for statistical significance using Gehan-Breslow-Wilcoxon test with Bonferroni's correction. Fisher's exact test was used to compare categorical variables. Statistical outliers were determined with a ROUT test, using a false-positive rate (Q) of 0.1%, and were removed from the behavioral time-series data sets.

## Acknowledgements

We thank the National BioResource Project (NBRP) Medaka for providing the BAC clone used in this study and Drs. Lino Tessarollo and Donald L Court for the DY380 bacteria. We also thank Dr. Shin-ichi Higashijima for technical advice on generating the transgenic construct; Dr. Guro K Sandvik for help with transgenic fish; Tatsuya Fukataki for performing the receptor activation assay; and Thomas Fleming for language editing of the revised manuscript.

## Additional information

### Funding

| Funder | Grant reference number | Author |
|---|---|---|
| Ministry of Education, Culture, Sports, Science, and Technology | 25132705 | Kataaki Okubo |
| Ministry of Education, Culture, Sports, Science and Technology | 17H06429 | Kataaki Okubo |
| Japan Society for the Promotion of Science | 16H04979 | Kataaki Okubo |
| Japan Society for the Promotion of Science | 19H03044 | Kataaki Okubo |
| Japan Society for the Promotion of Science | 12J07446 | Towako Hiraki-Kajiyama |
| RIKEN | Special Postdoctoral Researcher Program | Towako Hiraki-Kajiyama |

The funders had no role in study design, data collection and interpretation, or the decision to submit the work for publication.

### Author contributions

Towako Hiraki-Kajiyama, Conceptualization, Formal analysis, Funding acquisition, Validation, Investigation, Visualization, Methodology, Writing—original draft, Writing—review and editing; Junpei Yamashita, Keiko Yokoyama, Yukiko Kikuchi, Mikoto Nakajo, Formal analysis, Validation, Investigation, Methodology, Writing—review and editing; Daichi Miyazoe, Yuji Nishiike, Kaito Ishikawa, Kohei Hosono, Yukika Kawabata-Sakata, Satoshi Ansai, Masato Kinoshita, Investigation, Methodology, Writing—review and editing; Yoshitaka Nagahama, Conceptualization, Supervision, Writing—review and editing; Kataaki Okubo, Conceptualization, Formal analysis, Supervision, Funding acquisition, Validation, Investigation, Visualization, Methodology, Writing—original draft, Writing—review and editing

### Author ORCIDs

Towako Hiraki-Kajiyama https://orcid.org/0000-0001-6684-823X
Mikoto Nakajo https://orcid.org/0000-0003-0749-9257
Satoshi Ansai http://orcid.org/0000-0003-2683-0160
Kataaki Okubo https://orcid.org/0000-0002-4178-3094

### Ethics

Animal experimentation: All animal procedures were performed in accordance with the guidelines of the Institutional Animal Care and Use Committee of the University of Tokyo. The committee requests the submission of an animal-use protocol only for use of mammals, birds, and reptiles, in accordance with the Fundamental Guidelines for Proper Conduct of Animal Experiment and Related Activities in Academic Research Institutions under the jurisdiction of the Ministry of Education, Culture, Sports, Science and Technology of Japan (Ministry of Education, Culture, Sports, Science and Technology, Notice No. 71; June 1, 2006). Accordingly, we did not submit an animal-use protocol for this study, which used only teleost fish and thus did not require approval by the committee.

Decision letter and Author response
Decision letter https://doi.org/10.7554/eLife.39495.032
Author response https://doi.org/10.7554/eLife.39495.033

## Additional files

### Supplementary files

• Supplementary file 1. Abbreviations of brain and spinal cord regions and brain nuclei.
DOI: https://doi.org/10.7554/eLife.39495.025

• Supplementary file 2. Primers used in this study.
DOI: https://doi.org/10.7554/eLife.39495.026

• Supplementary file 3. Species names and GenBank accession numbers of the protein sequences used in this study.
DOI: https://doi.org/10.7554/eLife.39495.027

• Transparent reporting form
DOI: https://doi.org/10.7554/eLife.39495.028

### Data availability

Sequence data have been deposited in DDBJ/EMBL/GenBank with accession number LC375958.

The following dataset was generated:

| Author(s) | Year | Dataset title | Dataset URL | Database and Identifier |
|---|---|---|---|---|
| Kataaki Okubo | 2019 | *Oryzias latipes npbwr2* mRNA for neuropeptides B and W receptor 2 | https://www.ncbi.nlm.nih.gov/nuccore/LC375958 | NCBI Genbank, LC375958 |

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
