## [Decision Letter]

Thank you for submitting your article "Neuropeptide B mediates female sexual receptivity in medaka fish, acting in a female-specific but reversible manner" for consideration by *eLife*. Your article has been reviewed by two peer reviewers, and the evaluation has been overseen by a Reviewing Editor and a Senior Editor. The following individual involved in review of your submission has agreed to reveal his identity: Scott Juntti (Reviewer #3).

The reviewers have discussed the reviews with one another and the Reviewing Editor has drafted this decision to help you prepare a revised submission.

Summary:

Hiraki-Kajiyama et al. follow up work on the sexually dimorphic expression of NPB with a series of experiments to understand its gene regulation and directly test its role in sexual behaviors using reverse genetics. An outstanding problem in molecular biology has been the linking of hormone receptors to individual target genes. A second problem is in discovering genes in the pathways that drive mating behavior. This paper addresses both, and makes an attempt to link the two. They report that the two paralogs of *npb, nbpa* and *npbb*, display female specific expression in Vs/Vp and PMm/PMg region of the medaka brain. They found that this female specific expression depends on estrogen and an ERE site in the case of *npba*, and is responsive to altered sex hormone milieu. Furthermore, they uncovered that in mutants lacking either the ligands (Npba and Npbb) or the receptor (Npbwr2) are less receptive to male courtship. Overall, the manuscript is well written, the data is of high quality, and the conclusions are supported by the results. There are a few issues need to be addressed.

Essential revisions:

The decreased expression of *npba* in the ∆ERE mutants supports the hypothesis that ER acts directly on the ERE to activate *npba* expression. However, the conclusion may be strengthened if *npba* expression in the mutants is not or less responsive to exogenous estrogen treatment.

The slow kinetics of estrogen induction of *npba* and *npbb* expression in Vs/Vp and PMm/PMg regions does not agree with the model of direct ER regulation. Other permissive changes may be necessary.

Regarding the change in the number of *npba/npbb* expression neurons, the authors only considered changes in gene expression. Is it possible that some of changes are due to neuronal birth/death in Vs/Vp and PMm/PMg during induction of sex reversal or change of sex hormone milieu?

The crosses that gave rise to fish used in behavioral experiments are not spelled out, and so the relationships between controls and experimental fish are unknown to the reader. This may be problematic as genetic variation may not be properly controlled for. Ideally, the genotypes analyzed would be derived from the same crosses and share parents. This is particularly important in the context of gene editing experiments due to off-target effects. If the wild-type controls are not derived from the same crosses as the mutants, these may behave differently due to factors other than the mutations under study.

A similar critique of crosses applies to the ∆ERE fish; are WT controls their siblings? It's worrisome that the ∆ERE females in Figure 1E have similar Npba mRNA levels to the WT females in 1C, and one explanation could be genetic- or cohort-level variations.

The authors postulate that single Npb gene deletions are insufficient to reveal a behavioral phenotype due to functional redundancy. However, some limitations of the data depicted in Figure 6—figure supplement 2 make this conclusion tenuous. First, the questions of genetic variability listed above apply here too. Second-and perhaps relatedly-the variability in the data appears large. WT fish have dramatically differing values when comparing row A and B in Figure 6—figure supplement 2 and Figure 6. Finally, the sample size is only ~1/3 that of Figure 6. Therefore this experiment appears underpowered to conclude that the single mutations don't have an effect. A power analysis would be appropriate to show that there is enough statistical power given the variability.

Discussion, fifth paragraph and elsewhere. Do androgens inhibit Npb expression? Androgen/estrogen balance is a parsimonious explanation, but Npb mRNA decrease after KT treatment could also be secondary to a decrease in circulating estrogen levels (or decreased neural synthesis of estrogen). Would not a better way to test this be the simultaneous treatment with KT and E2?

As estradiol levels affect Npba expression, it is important to confirm that E2 levels are not changed in the ∆ERE fish. As Npba is expressed in the pituitary, it's not improbable that the decreased Npba expression is due to low estrogen. Furthermore, treating these animals with E2 to test whether the gene has become unresponsive to ER would be a nice demonstration.

To appropriately test latency to perform behavior, without removing any animals from analysis, one should use Kaplan-Meier plots (Jahn-Eimermacher, 2011) Statistical analysis of latency outcomes in behavioral experiments). This does not require any assumptions about data distributions and allows the inclusion of animals that do not perform a behavior.

---

## [Author Response]

Essential revisions:The decreased expression of npba in the ∆ERE mutants supports the hypothesis that ER acts directly on the ERE to activate npba expression. However, the conclusion may be strengthened if npba expression in the mutants is not or less responsive to exogenous estrogen treatment.

Following this comment, we have examined *npba* expression in the brain of ∆ERE mutant females that were sham-operated, ovariectomized, or ovariectomized and treated with E2 by real-time PCR. In ∆ERE mutant females, *npba* expression was significantly reduced by ovariectomy and restored by E2 treatment; however, E2 was not able to restore *npba* expression to the level present in wild-type females. These results further support the conclusion that estrogen directly activates the expression of *npba* via the ERE found in its promoter, while simultaneously suggesting that other mechanisms are also involved in this estrogen effect.

These data have been included in the revised manuscript as follows:

1) Results section: “These results demonstrate that estrogen elicits the female-specific expression of *npba*, at least in part, by direct transcriptional activation through the ERE present in the *npba* promoter.” has been replaced with “We also evaluated *npba* expression in the whole brain of ΔERE mutant females that were sham-operated, ovariectomized, or ovariectomized and treated with estradiol-17β (E2; the primary estrogen in vertebrates including teleosts) by real-time PCR. […] Taken together, these results demonstrate that estrogen elicits the female-specific expression of *npba* by direct transcriptional activation through the ERE present in the *npba* promoter, although other mechanisms are also likely involved in this estrogen effect.”. Following the addition of this description, “estradiol-17β (E2; the primary estrogen in vertebrates including teleosts)” has been replaced with “E2”.

2) Materials and methods section: “For the analysis of *npba* expression in ovariectomized and E2-treated ΔERE mutant females, a two-way ANOVA was conducted to test for main effects and interactions between genotype and treatment.” has been added.

3) Legend for Figure 1: “(G) Levels of *npba* expression in the whole brain of ΔERE (beige columns) and WT (blue columns) females that were sham-operated (Sham), ovariectomized (OVX), or ovariectomized and treated with estradiol-17β (OVX+E2) as determined by real-time PCR (n = 6 per group). There were significant main effects of genotype (F (1, 30) = 45.03, *p* < 0.0001) and treatment (F (2, 30) = 60.19, *p* < 0.0001) and a significant interaction between genotype and treatment (F (2, 30) = 9.944, *p* = 0.0005). *, *p* < 0.05; **, *p* < 0.01; ***, *p* < 0.001 (Bonferroni’s *post hoc* test).” has been added.

4) Figure 1: A new panel showing these results (panel G) has been added and the layout of the other panels has been modified.

The slow kinetics of estrogen induction of npba and npbb expression in Vs/Vp and PMm/PMg regions does not agree with the model of direct ER regulation. Other permissive changes may be necessary.

As this comment indicates, the estrogenic induction of *npba/npbb* expression in Vs/Vp and PMm/PMg of males occurred rather slowly. We have previously reported that ERs are not expressed at detectable levels in Vs/Vp and PMm/PMg of males and that treating males with estrogen induces ER expression therein (Hiraki et al., 2012). Considering these findings, it seems plausible that *npba/npbb* expression is dependent on ER expression and thus the response to estrogen is slower than expected. In support of this idea, we have obtained preliminary evidence that a cumulative increase in ER expression precedes the elevation of *npba/npbb* expression in Vs/Vp and PMm/PMg of estrogen-treated males. We are currently investigating the mechanism by which estrogen induces ER expression and will report our findings in a future paper.

We have included part of this discussion in the revised manuscript by adding the following paragraph in the Discussion section: “The estrogenic induction of *npba* expression in Vs/Vp and PMm/PMg of males showed a slower kinetic response than would be expected of a model of direct transcriptional activation. […] Considering these findings, it seems plausible that this estrogen-induced *npba* expression is not fully realized until sufficient ER expression is achieved.”

Regarding the change in the number of npba/npbb expression neurons, the authors only considered changes in gene expression. Is it possible that some of changes are due to neuronal birth/death in Vs/Vp and PMm/PMg during induction of sex reversal or change of sex hormone milieu?

While the present paper focuses on deciphering gene expression patterns in *npba/npbb*-expressing neurons, we have also been interested in this possibility, given that the brains of teleosts exhibit extensive adult neurogenesis and that estrogen has significant effects on adult neurogenesis. We have published a paper relevant to this issue recently (Kikuchi et al., 2019). In this paper, we found that Npb-expressing neurons emerged in adult male medaka treated with estrogen, not through neurogenesis, but rather through the activation of pre-existing, quiescent male counterpart neurons. This finding strongly suggests that an activation–inactivation process on existing neurons is regulating the number of *npba/npbb*-expressing neurons.

This discussion has been included in the revised manuscript as follows:

1) Discussion section: “The brains of teleosts exhibit extensive adult neurogenesis (Chapouton et al., 2007; Ganz and Brand, 2016) and estrogen has significant effects on adult neurogenesis (Mahmoud et al., 2016; Heberden, 2017; Ponti et al., 2018). […] This evidence suggests the relevance of an activation–inactivation process on existing neurons, rather than neurogenesis, in the reversible sexual dimorphism in *npba* expression.” has been added.

2) Reference list - the following references, which are cited in the above discussion, have been added:

Chapouton, Jagasia and Bally-Cuif, 2007; Ganz and Brand, 2016; Heberden, 2017; Kikuchi et al., 2019; Mahmoud, Wainwright and Galea, 2016; Ponti et al., 2018.

The crosses that gave rise to fish used in behavioral experiments are not spelled out, and so the relationships between controls and experimental fish are unknown to the reader. This may be problematic as genetic variation may not be properly controlled for. Ideally, the genotypes analyzed would be derived from the same crosses and share parents. This is particularly important in the context of gene editing experiments due to off-target effects. If the wild-type controls are not derived from the same crosses as the mutants, these may behave differently due to factors other than the mutations under study.

In the behavioral analysis of *npba* single knockouts, their wild-type and heterozygous siblings served as controls. Similarly, in the analysis of *npbb* single knockouts, their heterozygous siblings served as controls. However, their wild-type siblings could not be used because *npbb* is located on the sex chromosomes and it is therefore impossible to obtain wild-type siblings of *npbb* knockouts through the mating of any pair of genotypes. For this reason, non-sibling wild-type fish were used as controls in the analysis of *npba/npbb* double knockouts and *npbwr2* knockouts. Additionally, we believe there is sufficient reason to support the reliability of our results considering similar phenotypes were detected in *npba* single knockouts, *npba/npbb* double knockouts, and *npbwr2* knockouts; non-sibling wild-type and sibling heterozygous controls gave similar results in all analyzed parameters in *npbb* single knockouts; all knockouts and wild-type controls used in the study were derived from a d-rR strain maintained in a closed colony (sharing the same genetic background) in our laboratory for over 10 years; and all fish used for each experiment were reared under the same conditions to minimize the influence of environmental factors.

We have included these pieces of information in the Materials and methods section as follows: “In the behavioral analysis of *npba* single knockouts, their wild-type and heterozygous siblings served as controls. In the analysis of *npbb* single knockouts, their heterozygous siblings and non-sibling wild-type fish served as controls, because *npbb* is located on the sex chromosomes and it is impossible to obtain wild-type siblings of *npbb* knockouts by the mating of any pair of genotypes. […] These non-sibling wild-type fish were derived from the same genetic background and reared under the same conditions as the knockouts.” has been added.

A similar critique of crosses applies to the ∆ERE fish; are WT controls their siblings? It's worrisome that the ∆ERE females in Figure 1E have similar Npba mRNA levels to the WT females in 1C, and one explanation could be genetic- or cohort-level variations.

In the analyses of ∆ERE mutants, their wild-type siblings were used as controls. Therefore, the difference in the level of *npba* expression between Figures 1C and 1E cannot be attributed to differences in the genetic background of the controls used but is most likely due to variations in the physiological conditions.

To specify the genetic background of the control fish used in this study, the following sentence has been added to the Materials and methods section: “In all analyses of ∆ERE mutants, their wild-type siblings were used as controls.”

The authors postulate that single Npb gene deletions are insufficient to reveal a behavioral phenotype due to functional redundancy. However, some limitations of the data depicted in Figure 6—figure supplement 2 make this conclusion tenuous. First, the questions of genetic variability listed above apply here too. Second-and perhaps relatedly-the variability in the data appears large. WT fish have dramatically differing values when comparing row A and B in Figure 6—figure supplement 2 and Figure 6. Finally, the sample size is only ~1/3 that of Figure 6. Therefore this experiment appears underpowered to conclude that the single mutations don't have an effect. A power analysis would be appropriate to show that there is enough statistical power given the variability.

As described above for the behavioral analysis of *npba* single knockouts, their wild-type and heterozygous siblings served as controls. All knockout and wild-type fish used in the present study, including *npbb* knockouts and their wild-type controls, share the same genetic background since they were derived from the d-rR strain maintained in a closed colony in our laboratory for over 10 years. The possibility can therefore be eliminated that the large variability in the data was due to the genetic diversity of the fish analyzed. Additionally, all fish used for each analysis were reared under the same conditions to minimize the influence of environmental factors. These facts notwithstanding, a relatively large variability was seen in the data of *npba* and *npbb* single knockouts, which may have limited our ability to detect significant differences in the data sets, as pointed out in this comment. Indeed, *post hoc* power analysis on the latency data of *npba* and *npbb* single knockouts indicated that we had very low power to detect differences (statistical power = 0.17–0.22). Accordingly, we have substantially increased the sample size as follows:

1) The numbers of wild-type, heterozygous, and homozygous fish used in the analysis of *npba* knockouts have been increased from 11, 12, and 12 to 34, 33, and 31, respectively.

2) The numbers of wild-type, heterozygous, and homozygous fish used in the analysis of *npbb* knockouts have been increased from 9, 10, and 12 to 29, 33, and 32, respectively.

As expected, this increase in sample size greatly improved the statistical power of the analyses (0.46–0.54), which is comparable to the analysis of *npba/npbb* double knockouts and *npbwr2* knockouts (0.51–0.53).

To our surprise (but perhaps not to this reviewer’s), the increase in sample size has yielded significant differences in the latencies from the first courtship display to the first wrapping and to the wrapping that resulted in spawning between *npba*^+/+^ and *npba*^-/-^ and between *npba*^+/-^ and *npba*^-/-^ (please see Figure 6). This is important since the phenotypes observed in *npba/npbb* double knockouts and *npbwr2* knockouts have now also been observed in *npba* single knockouts. These results further strengthen the role of NPB signaling in female sexual receptivity. We deeply appreciate this discerning comment from this reviewer.

The text and figures have been revised to reflect these results as follows:

1) Results section: “Moreover, there was no significant difference in any of the detailed behavioral parameters among wild-type, heterozygous, and homozygous females of either knockout strain (Figure 6—figure supplement 2). […] However, subsequent quantitative behavioral testing showed abnormalities in several parameters.” has been changed to “However, subsequent quantitative behavioral testing uncovered a latent abnormality in *npba* knockout strain.”.

2) Results section: “Here, we found a significant increase in the latency from the first courtship display to the first wrapping for both *npba*^-/-^/*npbb*^-/-^ and *npbwr2*^-/-^ females as compared with wild-type females” has been changed to “Here, we found a significant increase in latencies from the first courtship display to the first wrapping and to the wrapping that resulted in spawning for *npba*^-/-^ females as compared with *npba*^+/+^ and *npba*^+/-^ females (*p* = 0.0256 and 0.0002, respectively, for the latency to the first wrapping; *p* = 0.0104 and 0.0026, respectively, for the latency to the wrapping that resulted in spawning) (Figure 6A). […] However, as was the case with *npba*^-/-^ females, a significant increase in the latency from the first courtship display to the first wrapping was observed for both *npba*^-/-^/*npbb*^-/-^ and *npbwr2*^-/-^ females as compared with wild-type females”.

3) Discussion section: “altered behavioral phenotypes were evident in *npba*^-/-^/*npbb*^-/-^ females” has been changed to “part of the altered behavioral phenotypes was evident in *npba*^-/-^/*npbb*^-/-^ females”.

4) Discussion section: “*npba*^-/-^/*npbb*^-/-^ females showed increased latency from courtship display to wrapping and increased incidence of spawning without any preceding courtship display. Consistent with this, *npbwr2*^-/-^ females showed identical phenotypes.” has been changed to “*npba*^-/-^, *npba*^-/-^/*npbb*^-/-^, and *npbwr2*^-/-^ females showed increased latency from courtship display to wrapping, and furthermore, *npba*^-/-^/*npbb*^-/-^ and *npbwr2*^-/-^ females showed increased incidence of spawning without any preceding courtship display.”.

5) Figure 6—figure supplement 2 has been revised to reflect the new results and renumbered as Figure 6 to be included in the main body of the paper.

6) As a consequence, Figure 6 has been renumbered as Figure 7. “Figure 6” has been replaced with “Figure 7A”.

7) Legend for new Figure 6 (previous Figure 6—figure supplement 2) now reads: “Figure 6. *npba* is involved in female sexual receptivity. Various parameters in the mating behavior of *npba* (A) and *npbb* (B) single knockout females were measured and compared with wild-type females (n = 34, 33, and 31 for *npba*^+/+^, *npba*^+/-^, and *npba*^-/-^ females, respectively; n = 29, 33, and 32 for *npbb*^+/+^, *npbb*^+/-^, and *npbb*^-/-^ females, respectively). Blue, ocher, and beige columns represent wild-type, heterozygous knockout, and homozygous knockout females, respectively. *, *p* < 0.05; **, *p* < 0.01; ***, *p* < 0.001 (Bonferroni’s *post hoc* test). See also Figure 6—figure supplement 1.”

8) Legend for new Figure 7 (previous Figure 6): “See also Figure 6—figure supplement 1 and Figure 6—figure supplement 2.” has been deleted.

Discussion, fifth paragraph and elsewhere. Do androgens inhibit Npb expression? Androgen/estrogen balance is a parsimonious explanation, but Npb mRNA decrease after KT treatment could also be secondary to a decrease in circulating estrogen levels (or decreased neural synthesis of estrogen). Would not a better way to test this be the simultaneous treatment with KT and E2?

To examine the possibility raised in this comment, we have measured E2 levels in the brains of KT-treated females. More specifically, females were treated with KT in exactly the same way as for the *npba/npbb* expression analysis and brain levels of E2 were measured on days 0, 2, 5, and 9 by LC-MS/MS. The results showed that brain levels of E2 fell to less than 5% of untreated control levels within 2 days and remained low during the remainder of the treatment period, demonstrating a substantial decrease in E2 levels in the brain of KT-treated females. This result strongly indicates that the decreased *npba/npbb* expression after KT treatment is secondary to a decrease in E2 levels, as suggested by this comment. We greatly appreciate this comment, which has much improved our understanding of the mechanisms underlying the reversal of sexually dimorphic *npba/npbb* expression.

We have included this data in the manuscript and revised the relevant discussion as follows:

1) Results section: “To aid in interpreting these results, we measured brain levels of E2 in females treated with KT in exactly the same way as for the *npba* expression analysis. The results showed that E2 levels fell to less than 5% of untreated controls within 2 days and remained low during the remainder of the treatment period (*p* < 0.0001 on days 2, 5, and 9), demonstrating a substantial decrease in E2 levels in the brain of KT-treated females (Figure 2—figure supplement 2).” has been added.

2) Discussion section: “Because ER expression in Vs/Vp and PMp/PMg is strongly suppressed by androgen (Hiraki et al., 2012), the androgen/AR signaling pathway presumably exerts its influence by attenuating the stimulatory action of estrogen through a reduction of ER in *npba*-expressing neurons. […] This represents an efficient system whereby brain and behavior are sexually differentiated, but the potential for sex reversal is retained.” has been replaced with “Considering a concomitant decrease in brain levels of E2 in androgen-treated males, the androgen/AR signaling pathway presumably exerts its influence by attenuating the stimulatory action of estrogen through a reduction of brain E2 levels. Given that ER expression in Vs/Vp and PMp/PMg is strongly suppressed by androgen (Hiraki et al., 2012), the inhibitory action of androgen on *npba* expression may involve repression of estrogen/ER signaling not only at ligand, but also at receptor level.”.

3) Materials and methods: “Measurement of brain levels of E2. Dissected whole brains were frozen and stored at -80ºC until analysis. Levels of E2 in the brains were analyzed by liquid chromatography coupled to tandem mass spectrometry (LC-MS/MS) at Aska Pharmamedical (Kanagawa, Japan). In the KT treatment experiment, E2 levels were measured over the same time course as that used to assess *npba/npbb* expression (on days 0 (untreated controls), 2, 5, and 9 of treatment).” has been added.

4) Legend for Figure 2: “See also Figure 2—figure supplement 1.” has been changed to “See also Figure 2—figure supplement 1 and Figure 2—figure supplement 2.”.

5) The legend for Figure 2—figure supplement 2 has been added. It reads: “Figure 2—figure supplement 2. Levels of estradiol-17β (E2) in the brains of females treated with 11-ketotestosterone (KT). n = 4 per sampling day. ***, *p* < 0.001 (versus day 0, Dunnett’s *post hoc* test).”

6) Figure 2—figure supplement 2 has been added.

As estradiol levels affect Npba expression, it is important to confirm that E2 levels are not changed in the ∆ERE fish. As Npba is expressed in the pituitary, it's not improbable that the decreased Npba expression is due to low estrogen. Furthermore, treating these animals with E2 to test whether the gene has become unresponsive to ER would be a nice demonstration.

In response to this comment, we have measured E2 levels in the brains of ∆ERE mutant females. The results showed that ∆ERE mutant females had E2 levels comparable to those of their wild-type sibling females, addressing the first concern in this comment.

This data has been included in the revised manuscript as follows:

1) Results section: “The possibility that decreased *npba* expression in ∆ERE mutant females was due to a decrease in estrogen levels in their brains was ruled out by the observation that ∆ERE mutant females had brain levels of E2 comparable to those of wild-type females (*p* = 0.782) (Figure 1—figure supplement 2).” has been added.

2) Legend for Figure 1: “See also Figure 1—figure supplement 1.” has been changed to “See also Figure 1—figure supplement 1 and Figure 1—figure supplement 2.”.

3) The legend for Figure 2—figure supplement 2 has been added. It reads: “Figure 1—figure supplement 2. Levels of estradiol-17β (E2) in the brains of ∆ERE mutant and wild-type (WT) females. n = 4 per group.”

4) Figure 1—figure supplement 2 has been added.

The second concern in this comment is the same as that in the first comment. As described in response to the first comment, we have examined *npba* expression in the brain of ∆ERE mutant females that were sham-operated, ovariectomized, or ovariectomized and treated with E2. The results showed that, in the mutant female brain, *npba* expression was significantly reduced by ovariectomy and restored by E2 treatment, but E2 was not effective to increase *npba* expression to the level in wild-type females.

To appropriately test latency to perform behavior, without removing any animals from analysis, one should use Kaplan-Meier plots (Jahn-Eimermacher, 2011) Statistical analysis of latency outcomes in behavioral experiments). This does not require any assumptions about data distributions and allows the inclusion of animals that do not perform a behavior.

Following this comment, the latency data were analyzed using Kaplan-Meier plots with the inclusion of fish that did not spawn within the test period. Differences between Kaplan-Meier curves were tested for statistical significance using Gehan-Breslow-Wilcoxon test with Bonferroni’s correction, according to the Prism 8 Statistics Guide (https://www.graphpad.com/guides/prism/8/statistics/index.htm). Fish that spawned without any courtship display were excluded from this analysis as before, because it was deemed inappropriate to treat these fish as either those that did not spawn within the test period or those that spawned “zero” seconds after the first courtship display.

Although the significant difference disappeared in the latency from the first courtship display to the wrapping that resulted in spawning between *npba*^+/-^ and *npba*^-/-^ fish, all other significant differences detected by ANOVA and *post hoc* test were also detected by Kaplan-Meier plots and Gehan-Breslow-Wilcoxon test, which did not affect the final conclusion of this study. The results have been summarized and presented in Figures 6 and 7. Since ANOVA/*post hoc* test and Kaplan-Meier plots/Gehan-Breslow-Wilcoxon test gave an inconsistent statistical result for the latency between *npba*^+/-^ and *npba*^-/-^ fish as described above, we would like to present both the Kaplan-Meier plots and the original bar graphs (showing the results of ANOVA/*post hoc* test), rather than to replace the bar graphs with the Kaplan-Meier plots.

Specifically, we have revised the text and figures as follows:

1) Results section: “Behavioral time-series data sets were further analyzed using Kaplan-Meier plots with the inclusion of fish that did not spawn within the test period. Although the significant difference disappeared in the latency from the first courtship display to the wrapping that resulted in spawning between *npba*^+/-^ and *npba*^-/-^, all other significant differences detected were also detected by this analysis, supporting the robustness of our results (the latency to the first wrapping; *p* = 0.0480 for *npba*^+/+^ versus *npba*^-/-^ females, *p* = 0.0291 for *npba*^+/-^ versus *npba*^-/-^ females, *p* = 0.0162 for wild-type versus *npba*^-/-^/*npbb*^-/-^ females, *p* < 0.0002 for wild-type versus *npbwr2*^-/-^ females: the latency to the wrapping that resulted in spawning; *p* = 0.0129 for *npba*^+/+^ versus *npba*^-/-^ females, *p* = 0.1158 for *npba*^+/-^ versus *npba*^-/-^ females, *p* = 0.1068 for wild-type versus *npba*^-/-^/*npbb*^-/-^ females, *p* = 0.0006 for wild-type versus *npbwr2*^-/-^ females) (Figure 6C, D and Figure 7B).” has been added.

2) Materials and methods: “Behavioral time-series data sets were further analyzed using Kaplan-Meier plots with the inclusion of fish that did not spawn within the test period, following Jahn-Eimermacher et al., 2011. Differences between Kaplan-Meier curves were tested for statistical significance using Gehan-Breslow-Wilcoxon test with Bonferroni’s correction.” has been added.

3) Reference list: the following reference, which is cited in the above text, has been added.

Jahn-Eimermacher, Lasarzik I, and Raber, 2011.

4) Legend for Figure 6: “(A, B)” has been added.

5) Legend for Figure 6: “(C, D) The latency data for *npba* (C) and *npbb* (D) single knockouts were further analyzed using Kaplan-Meier plots. Blue triangles, ocher diamonds, and beige circles represent wild-type, heterozygous knockout, and homozygous knockout females, respectively. *, *p* < 0.05 (Gehan-Breslow-Wilcoxon test with Bonferroni’s correction).” has been added.

6) Legend for Figure 7: “(A)” has been added.

7) Legend for Figure 7: “(B) The latency data were further analyzed using Kaplan-Meier plots. Blue triangles, ocher diamonds, and beige circles represent wild-type, *npba/npbb* double knockout, and *npbwr2* knockout females, respectively. *, *p* < 0.05; ***, *p* < 0.001 (Gehan-Breslow-Wilcoxon test with Bonferroni’s correction).” has been added.

8) Figures 6 and 7: The Kaplan-Meier plots for the latency data have been added in these figures (panels C and D in Figure 6 and panel B in Figure 7).